# Disentangling molecular alterations from water-content changes in the aging human brain using quantitative MRI

Shir Filo [1,3], Oshrat Shtangel[1,3], Noga Salamon[1], Adi Kol[1], Batsheva Weisinger[1], Sagiv Shifman [2] & Aviv A. Mezer [1]

It is an open question whether aging-related changes throughout the brain are driven by a common factor or result from several distinct molecular mechanisms. Quantitative magnetic resonance imaging (qMRI) provides biophysical parametric measurements allowing for non-invasive mapping of the aging human brain. However, qMRI measurements change in response to both molecular composition and water content. Here, we present a tissue relaxivity approach that disentangles these two tissue components and decodes molecular information from the MRI signal. Our approach enables us to reveal the molecular composition of lipid samples and predict lipidomics measurements of the brain. It produces unique molecular signatures across the brain, which are correlated with specific gene-expression profiles. We uncover region-specific molecular changes associated with brain aging. These changes are independent from other MRI aging markers. Our approach opens the door to a quantitative characterization of the biological sources for aging, that until now was possible only post-mortem.

[1] The Edmond and Lily Safra Center for Brain Sciences, The Hebrew University of Jerusalem, Jerusalem 9190401, Israel. [2] Department of Genetics, The Institute of Life Sciences, The Hebrew University of Jerusalem, Jerusalem 9190401, Israel. [3] These authors contributed equally: Shir Filo, Oshrat Shtangel. Correspondence and requests for materials should be addressed to A.A.M. (email: aviv.mezer@elsc.huji.ac.il)

The biology of the aging process is complex, and involves various physiological changes throughout cells and tissues[1]. One of the major changes is atrophy, which can be monitored by measuring macroscale brain volume reduction[1,2]. In some cases, atrophy can also be detected as localized microscale tissue loss reflected by increased water content[3]. This process is selective for specific brain regions and is thought to be correlated with cognitive decline in Alzheimer's disease[2,4,5]. In addition to atrophy, there are molecular changes associated with the aging of both the normal and pathological brain[5,6]. Specifically, lipidome changes are observed with age, and are associated with several neurological diseases[7–11].

It is an open question as to whether there are general principles that govern the aging process, or whether each system, tissue, or cell deteriorates with age for different reasons[12,13]. On one hand, the common-cause hypothesis proposes that different biological aging-related changes are the result of a single underlying factor[14,15]. This implies that various biomarkers of aging will be highly correlated[16]. On the other hand, the mosaic theory of aging suggests that there are several distinct aging mechanisms that have a heterogenous effect throughout the brain[12,13]. According to this latter view, combining different measurements of brain tissue is crucial in order to fully describe the state of the aging brain. To test these two competing hypotheses in the context of volumetric and molecular aging-related changes, it is essential to measure different biological aspects of brain tissue. Unfortunately, the molecular correlates of aging are not readily accessible by current in vivo imaging methods.

The main technique used for non-invasive mapping of the aging process in the human brain is magnetic resonance imaging (MRI)[2,17–19]. Advances in the field have led to the development of quantitative MRI (qMRI). This technique provides biophysical parametric measurements that are useful in the investigation and diagnosis of normal and abnormal aging[20–27]. qMRI parameters have been shown to be sensitive to the microenvironment of brain tissue and are therefore named in vivo histology[28–30]. Nevertheless, an important challenge in applying qMRI measurements is increasing their biological interpretability. It is common to assume that qMRI parameters are sensitive to the myelin fraction[20,23,30–33], yet any brain tissue including myelin is a mixture of multiple lipids and proteins. Moreover, since water protons serve as the source of the MRI signal, the sensitivity of qMRI parameters to different molecular microenvironments may be confounded by their sensitivity to the water content of the tissue[34,35]. We hypothesized that the changes observed with aging in MRI measurements[20,23,30–33,36] such as R1, R2, mean diffusivity (MD), and magnetization transfer saturation (MTsat)[37], could be due to a combination of an increase in water content at the expense of tissue loss, and molecular alterations in the tissue.

Here, we present a qMRI analysis that separately addresses the contribution of changes in molecular composition and water content to brain aging. Disentangling these two factors goes beyond the widely accepted "myelin hypothesis" by increasing the biological specificity of qMRI measurements to the molecular composition of the brain. For this purpose, we generalize the concept of relaxivity, which is defined as the dependency of MR relaxation parameters on the concentration of a contrast agent[38]. Instead of a contrast agent, our approach exploits the qMRI measurement of the local non-water fraction[39] to assess the relaxivity of the brain tissue itself. This approach allows us to decode the molecular composition from the MRI signal. In samples of known composition, our approach provides unique signatures for different brain lipids. In the live human brain, it produces unique molecular signatures for different brain regions. Moreover, these MRI signatures agree with post-mortem measurements of the brain lipid and macromolecular composition, as well as with specific gene-expression profiles. To further validate the sensitivity of the relaxivity signatures to molecular composition, we perform direct comparison of MRI and lipidomics on post-mortem brains. We exploit our approach for multidimensional characterization of aging-related changes that are associated with alterations in the molecular composition of the brain. Finally, we evaluate the spatial pattern of these changes throughout the brain, in order to compare the common-cause and the mosaic theories of aging in vivo.

## Results

**Different brain lipids have unique relaxivity signatures**. The aging process in the brain is accompanied by changes in the chemophysical composition, as well as by regional alterations in water content. In order to examine the separate pattern of these changes, we developed a model system. This system was based on lipid samples comprising common brain lipids (phosphatidylcholine, sphingomyelin, phosphatidylserine, phosphatidylcholine-cholesterol, and phosphatidylinositol-phosphatidylcholine)[7]. Using the model system, we tested whether accounting for the effect of the water content on qMRI parameters provides sensitivity to fine molecular details such as the head groups that distinguish different membrane phospholipids. The non-water fraction of the lipid samples can be estimated by the qMRI measurement of lipid and macromolecular tissue volume (MTV, for full glossary of terms see Supplementary Table 1)[39]. By varying the concentration of the lipid samples, we could alter their MTV and then examine the effect of this manipulation on qMRI parameters. The parameters we estimated for the lipid samples were R1, R2, and MTsat. The potential ambiguity in the biological interpretation of qMRI parameters is demonstrated in Fig. 1a. On one hand, samples with similar lipid composition can present different R1 measurements (Fig. 1a, points 1 & 2). On the other hand, scanning samples with different lipid compositions may result in similar R1 measurements (Fig. 1a, points 2 & 3). This ambiguity stems from the confounding effect of the water content on the MR relaxation properties.

We evaluated the dependency of different qMRI parameters on the non-water fraction estimated by MTV. This analysis revealed strong linear dependencies (median $R^2 = 0.74$, Fig. 1a, b and Supplementary Fig. 1a, b). These linear MTV dependencies change as a function of the lipid composition, reflecting the inherent relaxivity of the different lipids. We could therefore use the MTV derivatives of qMRI parameters ($\frac{dqMRI}{dMTV}$, i.e., the slope of the linear relationship between each qMRI parameter and MTV) as a measure that is sensitive to molecular composition. By accounting for the Multidimensional Dependency on MTV ("MDM") of several qMRI parameters, a unique MRI relaxivity signature was revealed for each lipid (Fig. 1c). This implies that the water-related ambiguity demonstrated in the inset of Fig. 1a can be removed by measuring the MTV dependencies (Fig. 1c). Creating mixtures of several lipids provided supportive evidence for the generality of our framework. Figure 1d and Supplementary Fig. 1c show that the qMRI measurements of a mixture can be predicted by summing the MTV dependencies of pure lipids (for further details see Supplementary Note 1 and Supplementary Fig. 2). Furthermore, we used this biophysical model to predict the lipid composition of a mixture from its MDM measurements (Fig. 1e). This model provided a good estimation of the sphingomyelin (Spg) and phosphatidylserine (PS) content ($R^2 > 0.64$) but failed to predict phosphatidylcholine (PtdCho) content (for further details see Supplementary Note 2). While lipids are considered to be a major source of the MRI signal in the brain[40–45], our approach can be applied to other compounds to reveal differences in the MRI signal between different proteins,

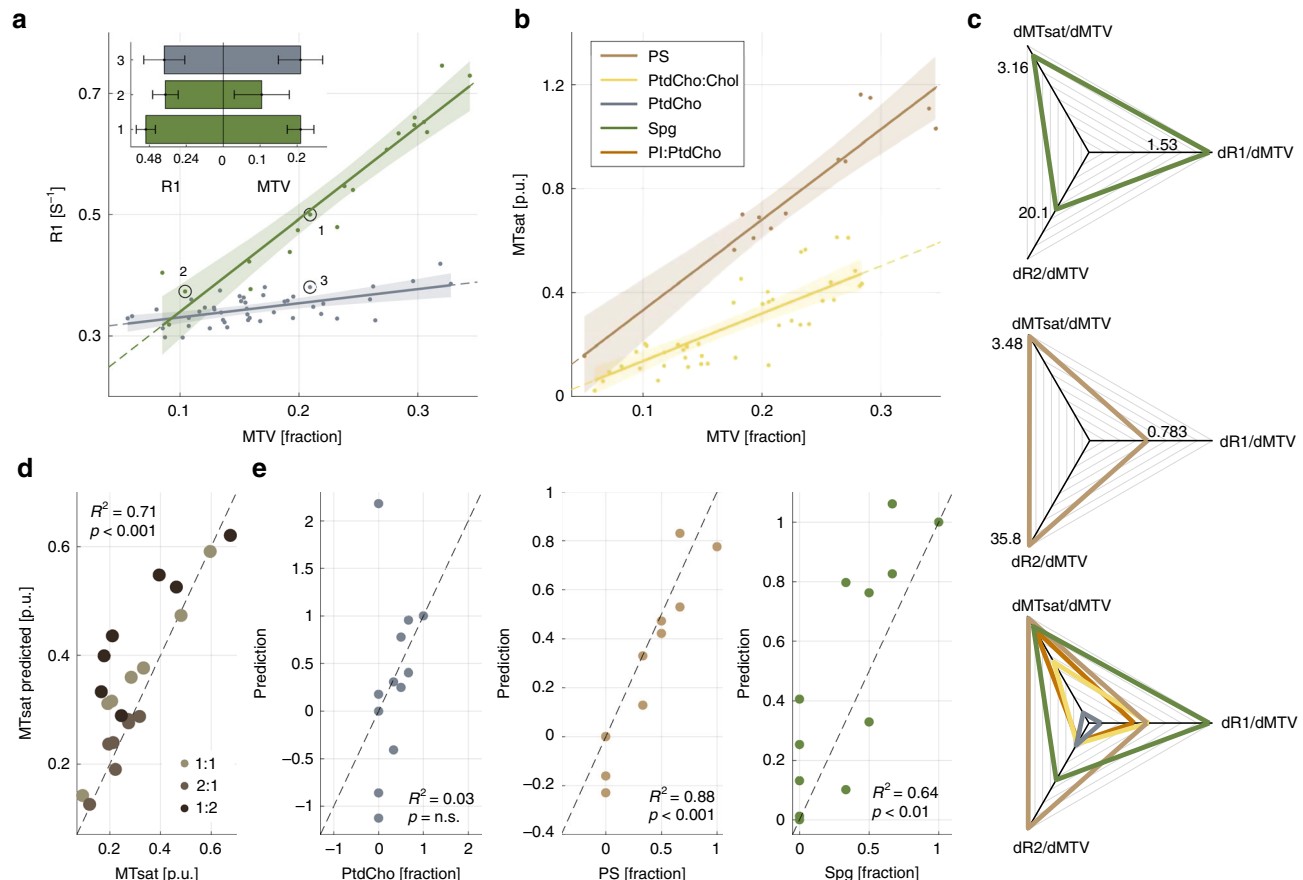

**Fig. 1** The dependency of qMRI parameters on the molecular composition in lipid samples. **a** The dependency of R1 on the lipid concentration (MTV) is shown for phosphatidylcholine (PtdCho) and sphingomyelin (Spg). The inset shows the ambiguity in the biological interpretation of R1. Bars represent R1 and MTV values of the three circled lipid samples; The same lipid at different concentrations can lead to different R1 (points 1 & 2), and two different lipids at different concentrations can have similar R1 (points 2 & 3). The main figure shows that the MTV derivative of R1 is specific for the lipid type; data points represent the median of lipids samples with varying concentrations. The linear relationships between R1 and MTV are marked by lines. Shaded areas represent the 95% confidence bounds. The slopes (MTV derivatives of R1) are different for each lipid. **b** The dependency of MTsat on the lipid concentration is shown for phosphatidylserine (PS) and phosphatidylcholine-cholesterol (PtdCho-Chol). **c** Unique MDM signatures of brain lipids. Each axis represents the MTV derivative of a different qMRI parameter (R1, MTsat, and R2). Colored traces extend between the MDM measurements of each lipid. Upper panels show individual lipids and an overlay of five lipids is in the lower panel (Spg, PtdCho, PS, PtdCho-Chol, and phosphatidylinositol-phosphatidylcholine (PI-PthCho)). **d** Predicting the MRI signal of lipid mixtures from the signal of pure lipids. MTsat measurements of PtdChol-PS mixtures (x-axis) can be predicted from a linear sum of the MTV dependencies of the pure lipids (y-axis). Different colors represent mixtures with different PtdCh-PS ratios. For each mixture we scanned samples with varying water concentrations. Dashed line is the identity line. **e** Predicting the lipid composition of 12 mixtures using the MDM method. The mixtures were composed of different PtdCho:PS:Spg ratios. MDM-based predictions were computed according to a biophysical model as a linear combination of the MDM measurements of the mixtures and the pure lipids. Predicted fractions of the three lipids (y-axes) are compared to their true fraction (x-axes). Dashed lines mark the identity line. p-values are for the F-test, n.s. = not significant

sugars, and ions (Supplementary Fig. 1d). Hence, the relationships between qMRI parameters and MTV account for the effect of water on MRI measurements and could be of use in quantifying the biological and molecular contributions to the MRI signal of water protons.

**The tissue relaxivity of the human brain is region-specific.** In order to target age-related changes in molecular composition, we applied the same approach for the human brain (Fig. 2a). We found that the linear dependency of qMRI parameters on MTV is not limited to in vitro samples and a similar relationship was also evident in the human brain (Fig. 2b and Supplementary Figs. 3–5). Importantly, different brain regions displayed a distinct dependency on MTV. Therefore, the relaxivity of brain tissue is region-specific. Figure 2b provides an example for the regional linear trends of R1 and MTsat in a single subject.

Remarkably, while the thalamus and the pallidum presented relatively similar R1 dependencies on MTV, their MTsat dependencies were different ($p < 0.001$, two-sample t-test). Compared to these two brain regions, frontal white-matter demonstrated different dependencies on MTV ($p < 0.001$, two-sample t-test). A better separation between brain regions can therefore be achieved by combining the MTV dependencies of several qMRI parameters (MTsat, MD, R1 and R2). The MTV derivatives of qMRI parameters are consistent across subjects (Fig. 2c and Supplementary Fig. 6), with good agreement between hemispheres (Supplementary Fig. 5). Moreover, they provide a novel pattern of differentiation between brain regions, which is not captured by conventional qMRI methods (Supplementary Fig. 7). In our lipid sample experiments, the MDM approach revealed unique relaxivity signatures of different lipids (Fig. 1c). Therefore, we attribute the observed diversity in the MTV derivatives of qMRI parameters across brain regions to the intrinsic heterogeneity in the

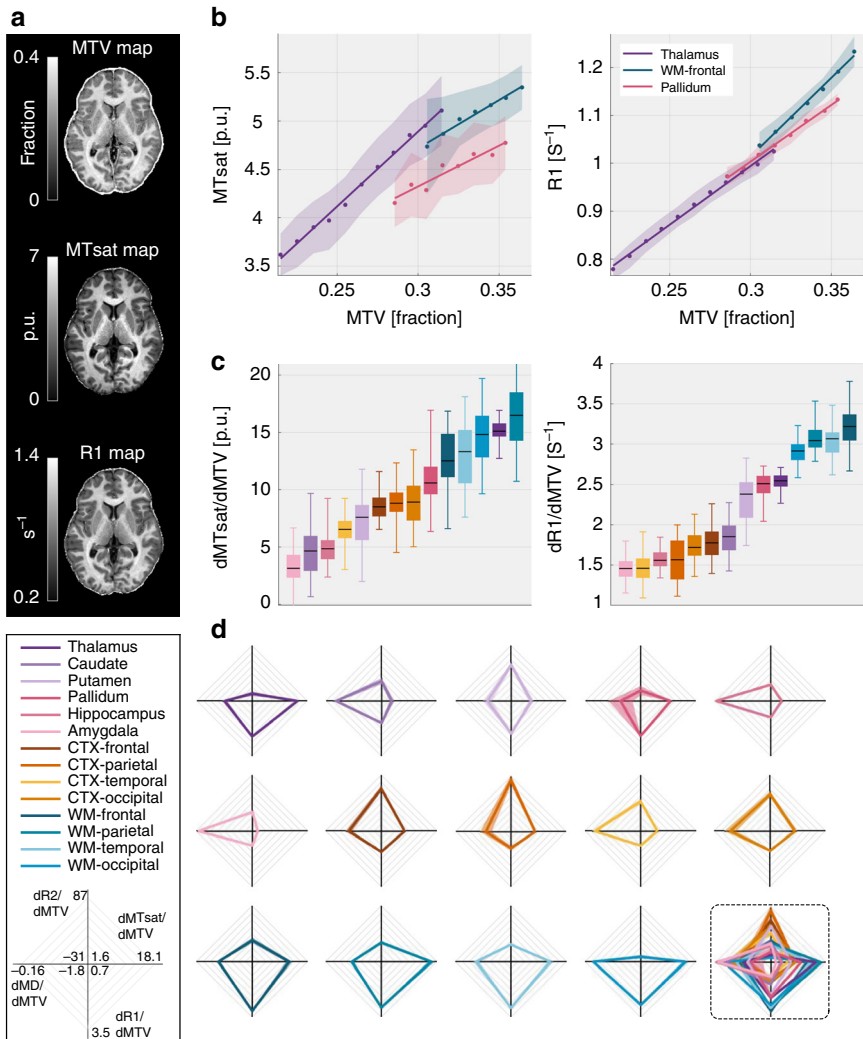

**Fig. 2** The MDM method provides region-specific signatures in the in vivo human brain. **a** Representative MTV, MTsat, and R1 maps. **b** Calculating the MDM signatures. The dependency of R1 (left) and MTsat (right) on MTV in three brain regions of a single subject. For each region, MTV values were pooled into bins (dots are the median of each bin; shaded area is the median absolute deviation), and a linear fit was calculated (colored lines). The slopes of the linear fit represent the MTV derivatives of R1 and MTsat and vary across brain regions. **c** The reliability of the MDM method across subjects. Variation in the MTV derivatives of R1 (left) and MTsat (right) in young subjects ($N = 23$). Different colors represent 14 brain regions (see legend). Edges of each box represent the 25th, and 75th percentiles, median is in black, and whiskers extends to extreme data points. Different brain regions show distinct MTV derivatives. **d** Unique MDM signatures for different brain regions (in different colors). Each axis is the MTV derivative ("MDM measurements") of a different qMRI parameter (R1, MTsat, R2, and MD). The range of each axis is in the legend. Colored traces extend between the MDM measurements, shaded areas represent the variation across subjects ($N = 23$). An overlay of all MDM signatures is marked with dashed lines

chemophysical microenvironment of these regions. The multi-dimensional dependency of various qMRI parameters on MTV can be represented by the space of MTV derivatives to reveal a unique chemophysical MDM signature for different brain regions (Fig. 2d, see explanatory scheme of the MDM method in Supplementary Fig. 8).

**The in vivo MDM approach captures ex vivo molecular profiles.** To validate that the MDM signatures relate to the chemophysical composition of brain tissue, we compared them to a previous study that reported the phospholipid composition of the human brain[7]. First, we established the comparability between the in vivo MRI measurements and the reported post-mortem data. MTV measures the non-water fraction of the tissue, a quantity that is directly related to the total phospholipid content. Indeed, we found good agreement between the in vivo measurement of

MTV and the total phospholipid content across brain regions ($R^2 = 0.95$, Fig. 3a). Söderberg et al.[7] identified a unique phospholipid composition for different brain regions along with diverse ratios of phospholipids to proteins and cholesterol. We compared this regional molecular variability to the regional variability in the MDM signatures. To capture the main axes of variation, we performed principal component analysis (PCA) on both the molecular composition of the different brain regions and on their MDM signatures. For each of these two analyses, the first principal component (PC) explained >45% of the variance. The regional projection on the first PC of ex vivo molecular composition was highly correlated ($R^2 = 0.84$, Fig. 3b) with the regional projection on the first PC of in vivo MDM signatures. This confirms that brain regions with a similar molecular composition have similar MDM. Supplementary Fig. 9a provides the correlations of individual lipids with MDM. Importantly, neither MTV nor the first PC of standard qMRI parameters was as strongly

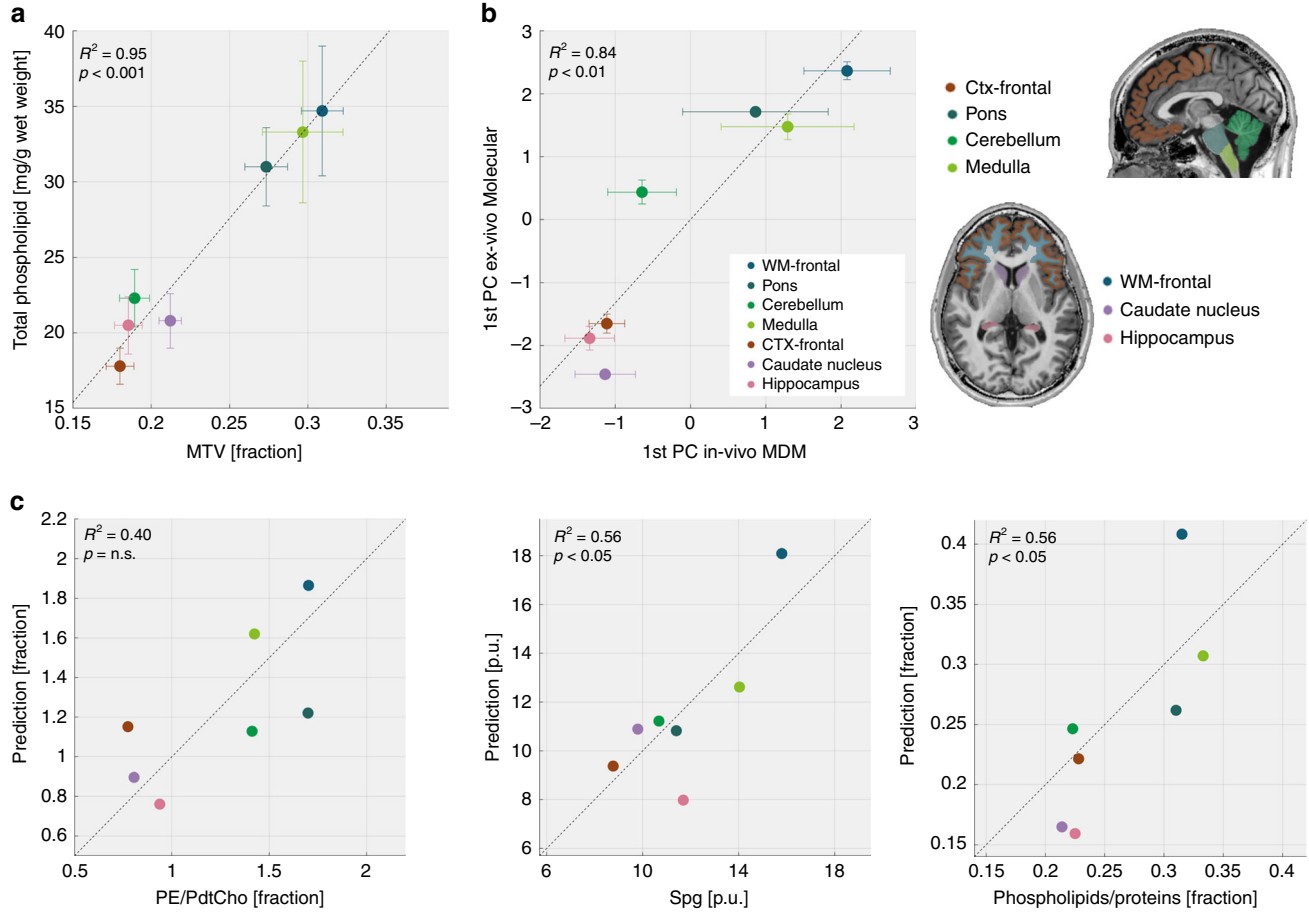

**Fig. 3** The biological interpretation of the MDM signatures based on comparison between in vivo and post-mortem data. Comparison of the in vivo MDM signatures of different brain regions to the molecular composition of these regions as reported in the literature for eight post-mortem human brains[7]. **a** Establishing the agreement between the post-mortem dataset and the in vivo MRI measurements. Comparison of the total phospholipid content derived from the literature ($N = 8$, y-axis) and the average MTV measurement across the young subjects ($N = 19$, x-axis) in seven different brain regions (colored data points). Adjusted $R^2$ values are presented for the entire figure. p-values are for the F-test in the entire figure. Error bars represent the standard deviation. **b** The similarity between the ex vivo molecular variability and the in vivo MDM variability across brain regions. The projection of different brain regions (colored data points, see legend on the right) on the 1st principal component (PC) of ex vivo molecular variability (y-axis, derived from the literature, $N = 8$) vs. their projection on the 1st principal component (PC) of in vivo MDM (x-axis, averaged over the young subjects, $N = 19$). PCs were computed across seven brain regions. The correlation between the two principal components indicates the similarity between the molecular and the MDM signatures. Error bars represent the standard deviation. **c** Predicting molecular composition with MRI. MDM-based prediction of different molecular features (y-axes, averaged over the young subjects, $N = 19$) compared to their true value (x-axes, $N = 8$) in different brain regions (colored data points, see legend on the right). The molecular features (PE/PtdCho ratio, Spg fraction, and ratio of phospholipids to proteins) were chosen as they account for most of the molecular variability across brain regions (as found in a PCA analysis). MDM-based predictions were computed from a weighted linear sum of the MTV derivatives of R1 and MTsat (as PCA indicates that they account for most of the variability in MDM across the brain). This linear model was fitted using leave-one-out cross validation

correlated with the ex vivo molecular composition as the MDM (Supplementary Fig. 9b, c). We next used the MDM measurements as predictors for molecular properties of different brain regions. Following our content predictions for lipids samples (Fig. 1e), we constructed a weighted linear model for human data (for further details see Supplementary Note 3). To avoid over fitting, we reduced the number of fitted parameters by including only the MDM and the molecular features that accounted for most of the regional variability. The MTV derivatives of R1 and MTsat accounted for most of the variance in MDM. Thus, we used these parameters as inputs to the linear model, while adjusting their weights through cross validation. We tested the performance of this model in predicting the three molecular features that account for most of the variance in the ex vivo molecular composition. Remarkably, MRI-driven MDM measurements provided good predictions for the regional

sphingomyelin composition ($R^2 = 0.56$, $p < 0.05$ for the F-test, Fig. 3c) and the regional ratio of phospholipids to proteins ($R^2 = 0.56$, $p < 0.05$ for the F-test, Fig. 3c).

Last, we compared the cortical MDM signatures to a gene co-expression network based on a widespread survey of gene expression in the human brain[46]. Nineteen modules were derived from the gene network, each comprised of a group of genes that co-varies in space. Six out of the nineteen gene modules were significantly correlated with the first PC of MDM. Interestingly, the first PC of MDM across the cortex was correlated most strongly with the two gene modules associated with membranes and synapses (Fig. 4, for further details see Supplementary Note 4 and Supplementary Figs. 10 and 11).

**Post-mortem validation for the lipidomic sensitivity of MDM.** The aforementioned analyses demonstrate strong agreement

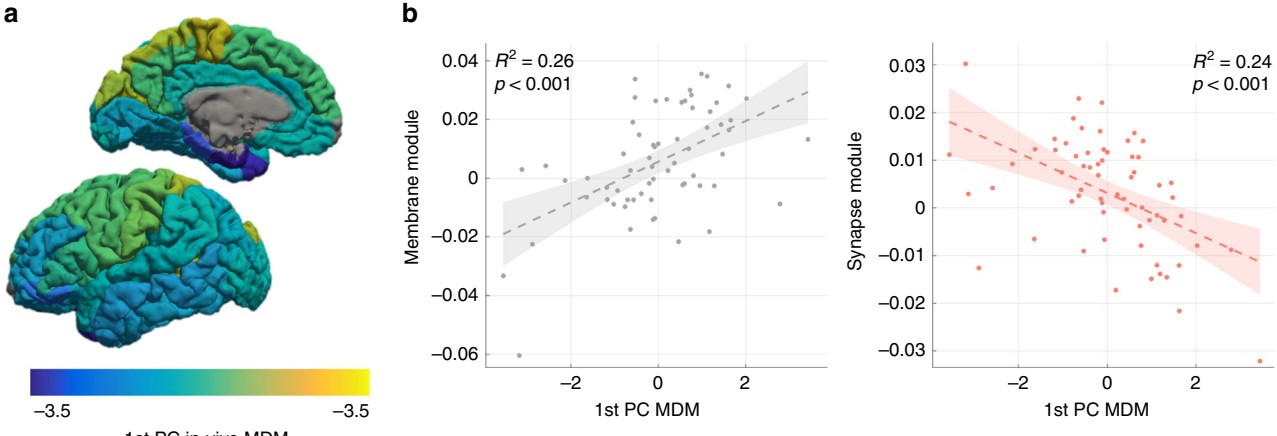

**Fig. 4** MDM correlation with specific gene-expression patterns throughout the cortex. **a** The projection of cortical areas on the 1st principal component (PC) of in vivo MDM is visualized on the cortical surface. MDM analysis was done on 64 cortical areas of the young subjects. In this case, the PCA is calculated using the R1 and MTsat maps (without MD and R2 maps), to allow sufficient resolution for cortical parcellation. **b** The correlation of the 1st PC of MDM with gene expression. Nineteen modules were derived from a gene co-expression network, each comprised of a group of genes that co-varies in space. The projection of different cortical areas on the first PC of each module captures the "eigengene" of the module. The two gene modules most correlated with MDM measurements are presented. The projection of 64 cortical areas on the eigengenes of the two gene modules (y-axes) is plotted against their projection on the 1st PC of in vivo MDM (x-axes, see **a**). The eigengenes represent the variability in the gene expression through the cortex. This variability correlates with the variability in MDM signatures. Shaded areas represent the 95% confidence bounds. Adjusted $R^2$ values presented. p-values are for the F-test and were corrected for multiple (57) comparisons. The analysis was done on the young subjects. The synapse module is termed "salmon" in Ben-David and Shifman[46], while the membrane module is termed "grey60"

between in vivo MDM measurements and ex vivo molecular composition based on a group-level comparison of two different datasets. Strikingly, we were able to replicate this result at the level of the single brain. To achieve this we performed MRI scans (R1, MTsat, R2, MD, and MTV mapping) followed by histology of two fresh post-mortem porcine brains (Fig. 5a, b). First, we validated the qMRI estimation of MTV using dehydration techniques. MTV values estimated using MRI were in agreement with the non-water fraction found histologically (adjusted $R^2 = 0.64$, $p < 0.001$ for the F-test, Fig. 5c).

Next, we estimated the lipid composition of different brain regions. Thin-layer chromatography (TLC) was employed to quantify seven neutral and polar lipids (Supplementary Table 2 and Supplementary Fig. 12a). In accordance with the analysis in Fig. 3, we performed PCA to capture the main axes of variation in lipidomics, standard qMRI parameters, and MDM. Figure 5d shows that MTV did not correlate with the molecular variability across the brain, estimated by the 1st PC of lipidomics. Likewise, the molecular variability did not agree with the 1st PC of standard qMRI parameters (Fig. 5e).

Last, we applied the MDM approach to the post-mortem porcine brain. Similar to the human brain, different porcine brain regions have unique MDM signatures (Fig. 5f, g and Supplementary Fig. 12b). Remarkably, we found that agreement between lipid composition and MRI measurements emerges at the level of the MDM signatures. The molecular variability across brain regions significantly correlated with the regional variability in the MDM signatures (adjusted $R^2 = 0.3$, $p < 0.01$ for the F-test, Fig. 5h). Excluding from the linear regression five outlier brain regions where the histological lipidomics results were 1.5 standard deviations away from the center yielded an even stronger correlation between MDM signatures and lipid composition (adjusted $R^2 = 0.55$, $p < 0.001$ for the F-test, Supplementary Fig. 12c). This post-mortem analysis validates that the MDM approach allows us to capture molecular information using MRI at the level of the individual brain.

**Disentangling water and molecular aging-related changes.** After establishing the sensitivity of the MDM signatures to the molecular composition of the brain, we used them to evaluate the chemophysical changes of the aging process. To assess aging-related changes across the brain, we scanned younger and older subjects (18 older adults aged 67 ± 6 years and 23 younger adults aged 27 ± 2 years). First, we identified significant molecular aging-related changes in the MDM signatures of different brain regions (Figs. 6 and 7, right column; Supplementary Fig. 13). Next, we tested whether the changes in MRI measurements, observed with aging, result from a combination of changes in the molecular composition of the tissue and its water content. We found that although it is common to attribute age-related changes in R1 and MTsat to myelin[28,30,36], these qMRI parameters combine several physiological aging aspects. For example, using R1 and MTsat we identified significant aging-related changes in the parietal cortex, the thalamus, the parietal white-matter and the temporal white-matter (Figs. 6 and 7, left column). However, the MDM approach revealed that these changes have different biological sources (Figs. 6 and 7, middle columns; see Supplementary Figs. 14–17 for more brain regions).

We discovered that the decrease in R1 values in the thalamus and parietal white-matter can be separated to an aging-related decrease in tissue volume (increase in water fraction), as estimated by MTV, and a strong chemophysical aging effect in the MTV derivative of R1 (Fig. 6a, b). In other brain regions there is a single biological source that generates most of the aging affect. For example, in the parietal cortex MTV values remained stable with age while the MTV derivative of R1 changed significantly (Fig. 6c). These findings suggest that the aging-related changes in R1 values in the parietal cortex result mainly from chemophysical alterations. On the other hand, in the temporal white-matter we did not find a significant aging-related change in the MTV derivative of MTsat (Fig. 7c). Therefore, the decrease in MTsat values with age in the temporal white-matter can be attributed mostly to a decrease in tissue volume, estimated by MTV (Fig. 7c).

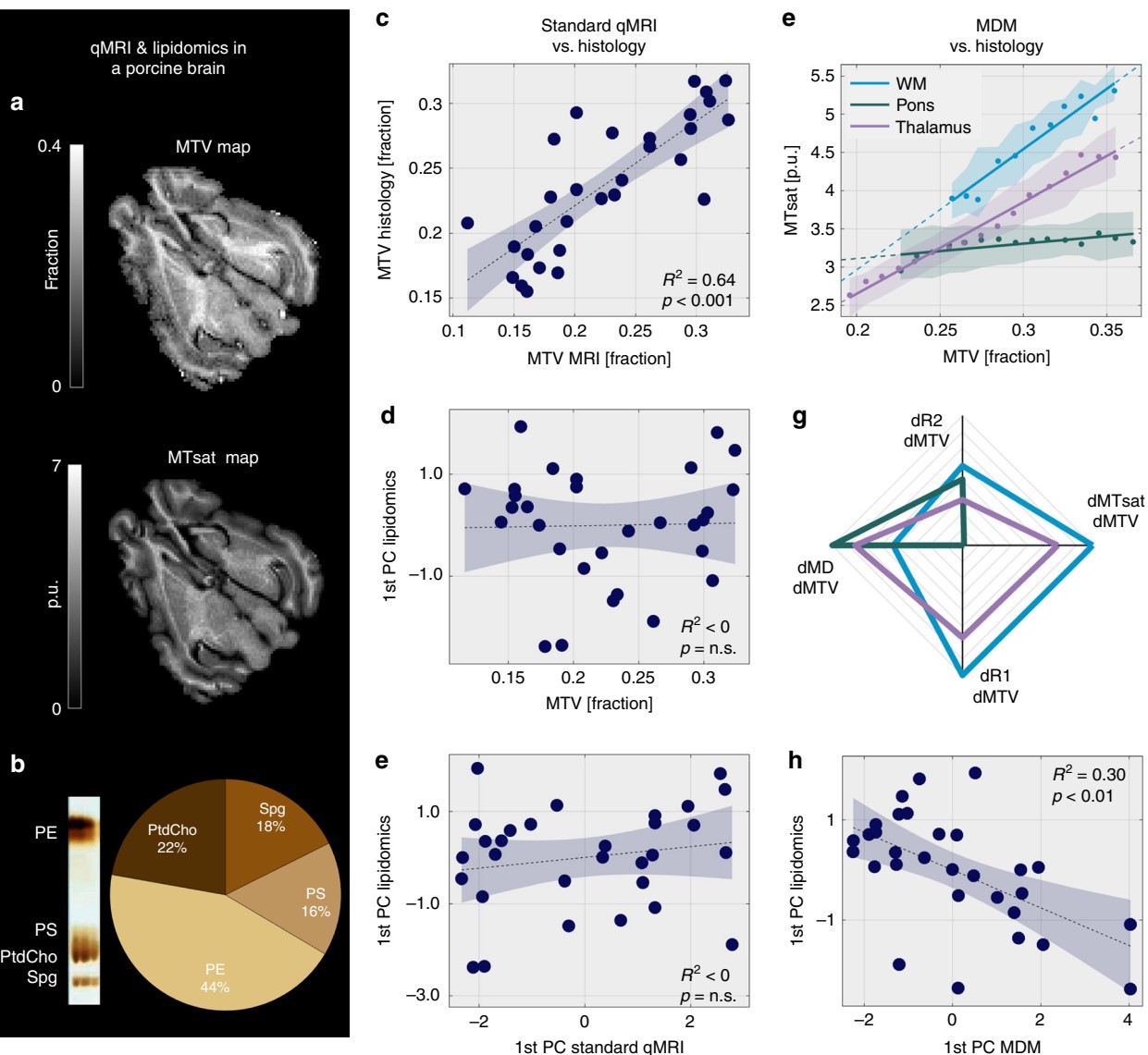

**Fig. 5** Post-mortem validation of the MDM approach. **a** Representative MTV and MTsat maps of a post-mortem poricne brain. **b** Lipidomics of the porcine brain. Left: representative column in a thin-layer chromatography (TLC) plate showing the phospholipid composition of a single brain sample. Neutral lipids are not shown. Right: The average percentage of the major four phospholipids (calculated by taking the median of each phospholipid over 30 samples). **c** Validation of the qMRI estimation of MTV. Comparison of the MTV measured using ex vivo evaporation technique (*y*-axis) and the MTV measured using MRI (*x*-axis) in 30 different brain regions. Shaded areas represent the 95% confidence bounds, adjusted $R^2$ values and *p*-values for the *F*-test are presented (also in **d**, **e**, **h**). **d**, **e** Standard qMRI parameters do not explain the molecular variability. The projection of 30 brain regions on the 1st principal component (PC) of lipidomics variability (*y*-axis) compared to MTV (**d**, *x*-axis) and to the projection of brain regions on the 1st PC of standard qMRI parameters (**e**, *x*-axis). **f–g** Post-mortem application of the MDM method. **f** The dependency of MTsat on MTV in three example brain regions. For each region, MTV values were pooled into bins (dots are the median of each bin; shaded area is the median absolute deviation), and a linear fit was calculated. The slopes of the linear fit (the MTV derivatives of MTsat) vary across brain regions. **g** Unique MDM signatures for different brain regions. Each axis is the MTV derivative of a different qMRI parameter (R1, MTsat, R2, and MD). The range of each axis was determined based on the most extreme data points. **h** The MDM signatures explain the molecular variability across brain regions. The projection of 30 brain regions on the 1st PC of lipidomics variability (*y*-axis, derived from TLC) vs. their projection on the 1st PC of MDM (*x*-axis). The correlation between the two principal components indicates the similarity between the molecular and the MDM signatures. Excluding from the linear regression five outlier brain regions with extreme TLC values yielded even stronger correlation between MDM signatures and molecular composition ($R^2 = 0.55$, Supplementary Fig. 12c)

Remarkably, in other brain regions the MDM approach revealed aging effects that are not captured by conventional qMRI methods. For example, MTsat measurements in the frontal cortex showed no significant differences between young and older individuals (Fig. 7a). However, by separating the water and chemophysical related effects of MTsat, as measured by MTV and MDM, respectively, we were able to identify significant changes between the age groups (Fig. 7a). In Fig. 3b we showed that the

1st PC of MDM increases the sensitivity of MRI to molecular composition relative to the 1st PC of standard qMRI parameters (Supplementary Fig. 9b, c). Interestingly, the 1st PC of MDM also reveals aging-related changes not captured by the 1st PC of standard qMRI parameters and MTV (Supplementary Fig. 18).

Iron is known to accumulate in different brain regions during aging, and in a variety of CNS disorders[47]. We investigated the contribution of iron to our findings using the estimation of R2*[48]

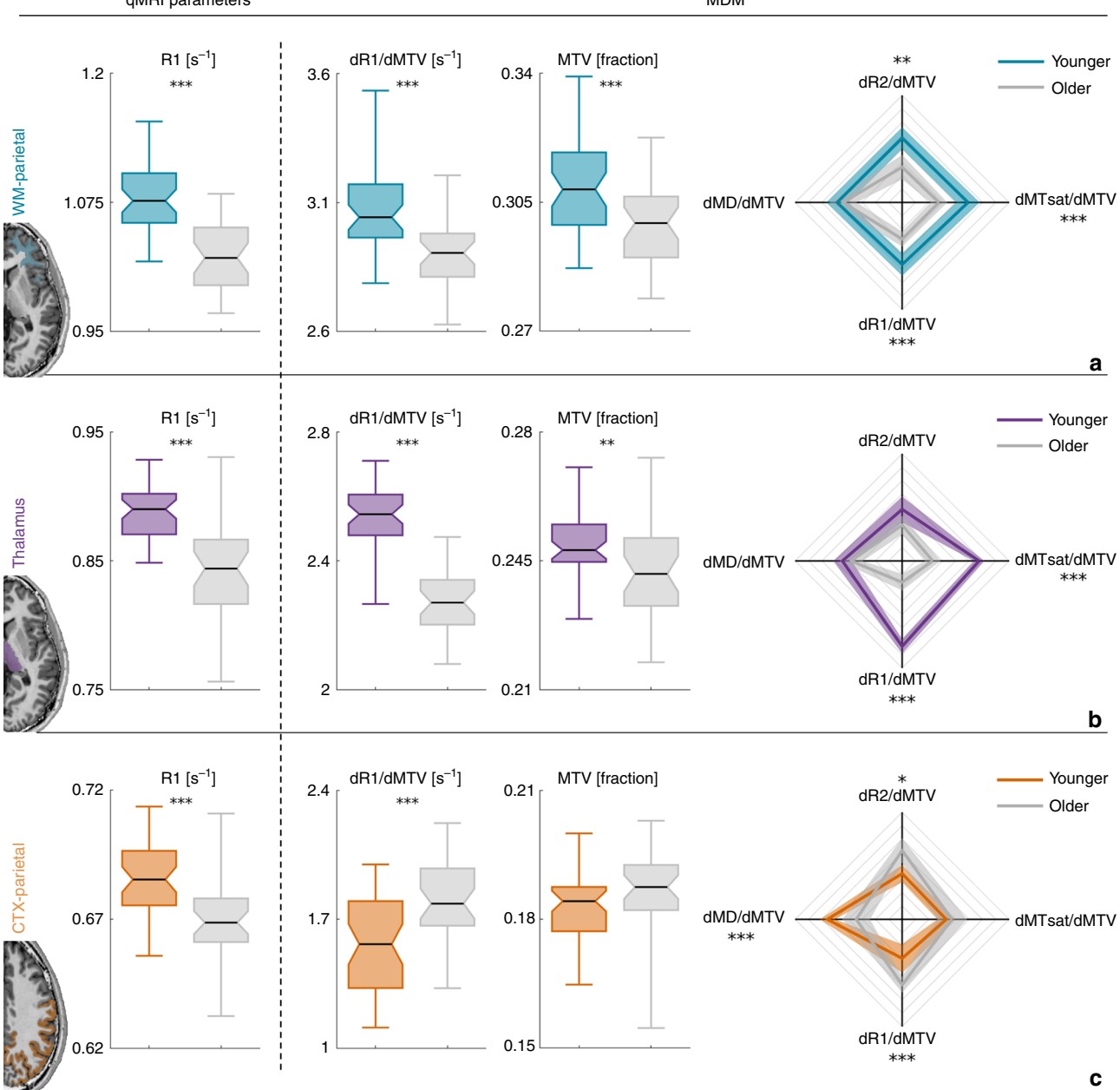

**Fig. 6** Region-specific aging-related molecular changes revealed by the R1 dependency on MTV. Comparison of MRI-driven measurements of 18 older adults (aged 67 ± 6 years, marked in gray) and 23 younger adults (aged 27 ± 2 years, marked with different colors) in the parietal white-matter (**a**), the thalamus (**b**), and the parietal cortex (**c**). Aging-related changes revealed by R1 are presented in the left column. The separate chemophysical and water-related contributions estimated by the MTV derivative of R1 and MTV, respectively, are shown in the middle columns. For each box, the central mark is the median, the box extends vertically between the 25th and 75th percentiles, the whiskers extend to the most extreme data points. Multidimensional aging-related changes revealed by the MDM approach are presented in the right column. Each axis represents the MTV derivative of a different qMRI parameter. Axes limits were set to the 5 and 95 percentiles. Traces extends between these derivatives, shaded areas represent the variation across subjects. The statistical significance of the differences between the groups was estimated using a two-sample *t*-test and was corrected for multiple comparisons using the FDR method. *$p < 0.05$; **$p < 0.01$; ***$p < 0.001$

(see Supplementary Note 5–6 and Supplementary Figs. 19–27). While we could detect R2*-related changes with age, we found that in most cases they could not explain the differences revealed by MDM measurements. Instead, the estimation of iron-related changes with age provides complementary information regarding the physiological modifications the brain tissue undergoes during aging. In addition, a biophysical model for the linear relationship between R1 and R2 to the inverse of the water content was suggested previously[43]. For alternative analysis of the qMRI

dependencies on to the inverse of the water content see Supplementary Fig. 28.

**Supporting evidence for the mosaic nature of brain aging**. Finally, we tested the common-cause and mosaic theories of aging in vivo. For this aim we compared the spatial patterns of different microscale and macroscale aging-related changes (Fig. 8a). Utilizing the MDM approach, we were able to portray the chemophysical aging trajectory throughout the brain

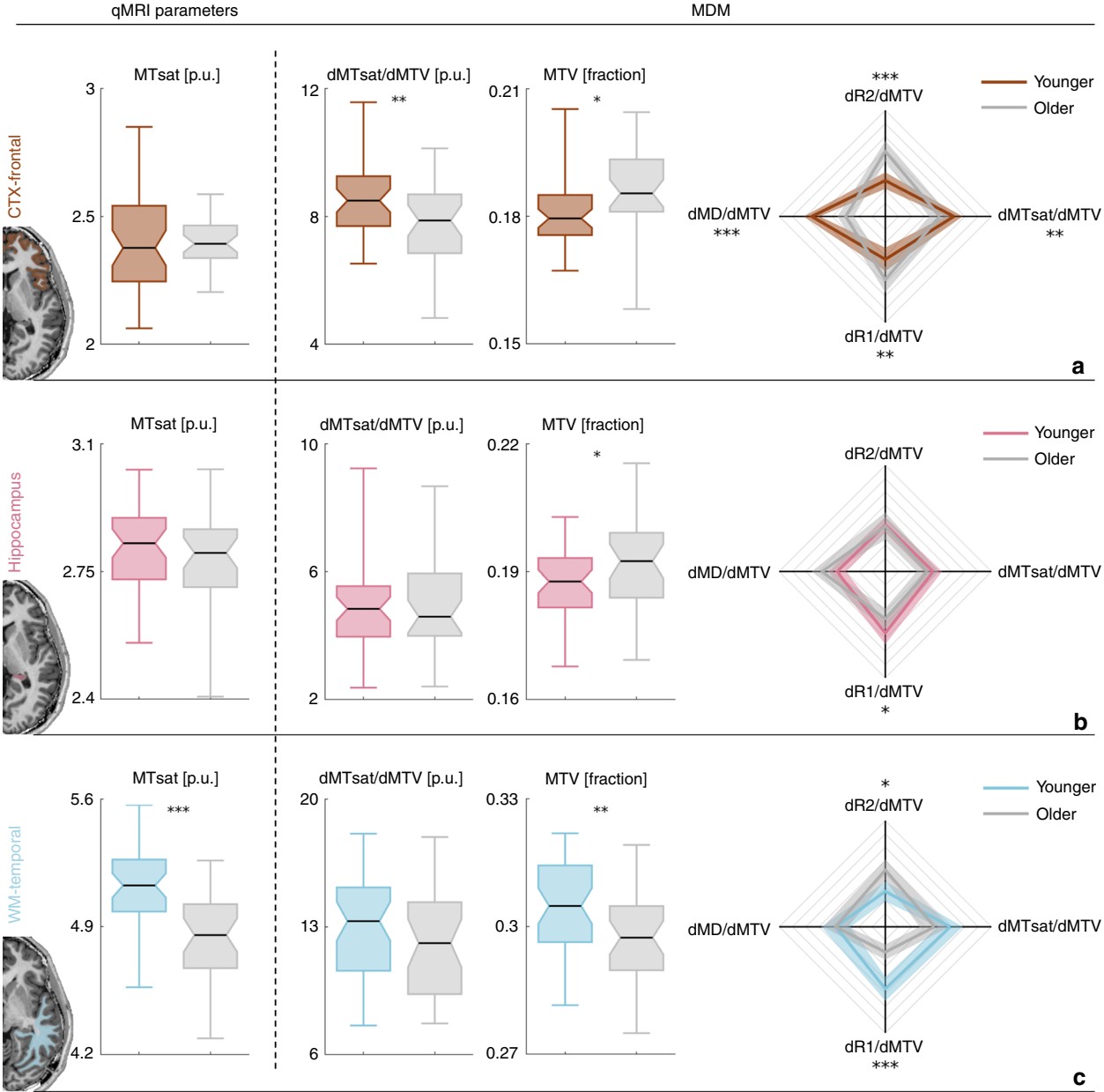

**Fig. 7** Region-specific aging-related molecular changes revealed by the MTsat dependency on MTV. Comparison of MRI-driven measurements of 18 older adults (aged 67 ± 6 years, marked in gray) and 23 younger adults (aged 27 ± 2 years, marked with different colors) in the frontal cortex (**a**), the hippocampus (**b**), and the temporal white-matter (**c**). Aging-related changes revealed by MTsat are presented in the left column. The separate chemophysical and water-related contributions estimated by the MTV derivative of MTsat and MTV, respectively, are shown in the middle columns. For each box, the central mark is the median, the box extends vertically between the 25th and 75th percentiles, the whiskers extend to the most extreme data points. Multidimensional aging-related changes revealed by the MDM approach are presented in the right column. Each axis represents the MTV derivative of a different qMRI parameter. Axes limits were set to the 5 and 95 percentiles. Traces extends between these derivatives, shaded areas represent the variation across subjects. The statistical significance of the differences between the groups was estimated using a two-sample $t$-test and was corrected for multiple comparisons using the FDR method. *$p < 0.05$; **$p < 0.01$; ***$p < 0.001$

(represented in Fig. 8a by the MTV derivative of MTsat, for other MDM dimensions see Supplementary Fig. 29). MTV was used to quantify the water-related trajectory[39], and R2* was employed to delineate the spatial pattern of aging-related changes in the iron concentration[48]. These trajectories reveal microscale aging-related pathways, yet macroscale alterations also occur with age. A widely studied macroscale characteristic of brain aging is atrophy, which can be estimated using MRI by measuring the total volume of different brain regions[1,2]. We compared the spatial trajectory of this macroscale property to

our microscale measurements of the human brain aging process (Fig. 8a). This analysis revealed that the aging of different brain regions is driven by different biological sources. Water-related changes are more substantial in white-matter regions, while changes in the iron content and volume with age are more pronounced in cortical regions. The chemophysical changes vary in space in a unique pattern compared to the other markers. They characterize the aging of cortical and white-matter regions, along with several sub-cortical structures, with a very large effect in the thalamus ($d > 2$).

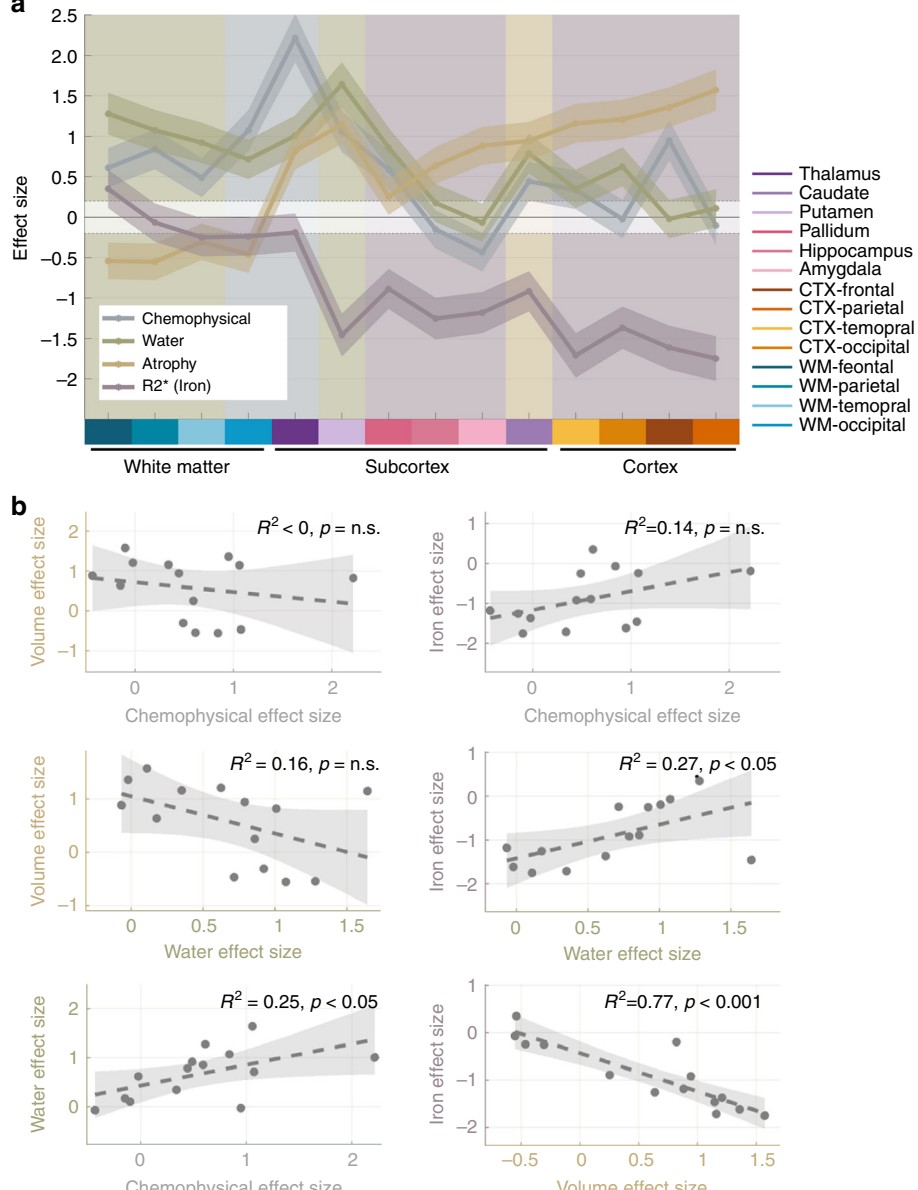

**Fig. 8** The "mosaic" nature of molecular and volumetric aging trajectories. **a** Distinct spatial patterns of different aging markers throughout the brain, in agreement with the mosaic theory. Different colors represent four markers of brain aging. The chemophysical, water and iron markers were estimated using dMTsat/dMTV, MTV, and R2*, respectively. The x-axis represent different brain regions (see legend). The y-axis shows the effect size of age-related changes between the younger ($N = 17$) and older ($N = 18$) groups evaluated using Cohen's d. Shaded areas mark the effect size's standard deviation. Background colors indicate which of the four markers has the largest effect size for each brain region. The dashed line marks the limit of small effect size ($d = 0.2$). dMTsat/dMTV and MTV were corrected for R2* effects. **b** The spatial correlation between aging-related changes in different biological markers. As expected from the mosaic theory, most of the different aging markers are not strongly correlated. Each plot shows the correlation in the effect sizes of two aging markers across different brain regions. Shaded areas represent the 95% confidence bounds for the fitted linear model. Adjusted $R^2$ values and p-values for the F-test are presented. n.s = not significant

In agreement with the mosaic hypothesis, we identified distinct aging patterns for different brain regions. For example, in the hippocampus we found a change in R2* values related to a higher iron concentration with age, along with significant reduction in the total hippocampal volume (Fig. 8a). This age-related shrinkage was not accompanied by lower MTV values, indicating conserved tissue density (Fig. 7b). In addition, there was no significant difference in the hippocampal MDM signature with age (Fig. 7b). Cortical gray-matter areas also exhibited similar trends of volume reduction without major loss in tissue density (Fig. 8a). Unlike the gray matter, in the white matter we did not

find volume reduction or large iron accumulation with age (Fig. 8a). However, we did find microscale changes with age in tissue composition, as captured by the MDM signature (Figs. 6a and 7c, and Supplementary Fig. 13), accompanied by a significant density-related decline in MTV (Fig. 8a). These findings are consistent with previous histological studies[49–51] (see Discussion), and provide the ability to monitor in vivo the different components of the aging mosaic.

Last, to test whether the different biological aging trajectories presented in Fig. 8a share a common cause, we evaluated the correlations between them (Fig. 8b). Importantly, the

chemophysical trajectory did not correlate significantly with the iron or volume aging patterns. The spatial distribution of water-related changes was found to correlate with iron content alterations ($R^2 = 0.27$) and chemophysical alterations ($R^2 = 0.25$). However, the strongest correlation between aging-related changes was found in volume and iron content ($R^2 = 0.77$). As shown previously, this correlation may be explained to some extent by a systematic bias in automated tissue classification[23]. Additional analysis revealed that the different dimensions of the MDM signature capture distinct patterns of aging-related changes (Supplementary Fig. 30). Hence, complementary information regarding the various chemophysical mechanisms underlying brain aging could be gained by combining them.

## Discussion

Normal brain aging involves multiple changes, at both the microscale and macroscale level. MRI is the main tool for in vivo evaluation of such age-related changes in the human brain. Here, we propose to improve the interpretation of MRI findings by accounting for the fundamental effect of the water content on the imaging parameters. This approach allows for non-invasive mapping of the molecular composition in the aging human brain.

Our work is part of a major paradigm shift in the field of MRI toward in vivo histology[30,36,52]. The MDM approach contributes to this important change by providing a hypothesis-driven bio-physical framework that was rigorously developed. We demonstrated the power of our framework, starting from simple pure lipid phantoms to more complicated lipid mixtures, and from there, to the full complexity of the brain. In the brain, we show both in vivo and post-mortem validations for the molecular sensitivity of the MDM signatures. Early observations relate different qMRI parameters to changes in the fraction of myelin[20,23,30–33,36]. The current approach enriches this view and provides better sensitivity to the molecular composition and fraction of myelin and other cellular tissues.

We developed a unique phantom system of lipid samples to validate our method. While the phantom system is clearly far from the complexity of brain tissue, its simplicity allowed us to verify the specificity of our method to the chemophysical environment. Remarkably, our approach revealed unique signatures for different lipids, and is therefore sensitive even to relatively subtle details that distinguish one lipid from another. We chose to validate our approach using membrane lipids based on previous experiments[40–45]. Nevertheless, we do acknowledge the fact that brain tissue comprises many other compounds beside lipids, such as proteins, sugars, and ions. As we have shown, these other compounds also exhibit unique dependency on MTV. The effect of such compounds, along with other factors such as micro-structure, and multi-compartment organization[28] is probably captured when we apply the MDM approach to the in vivo human brain. Therefore, the phantoms were made to examine the MRI sensitivity for the chemophysical environment, and the human brain data was used to measure the true biological effects in a complex in vivo environment.

Our relaxivity approach captures the molecular signatures of the tissue, but is limited in its abilities to describe the full complexity of the chemophysical environment of the human brain. For example, R1 and R2, which are used to generate the MDM signatures, are also sensitive to the iron content[23,48,52]. However, we found that most of our findings cannot be attributed to alterations in iron content as measured with R2* (for more details see Supplementary Note 5). While there is great importance in further isolating different molecular components, we argue that accounting for the major effect of water on qMRI parameters (for

$R^2$ distributions see Supplementary Fig. 5) is a crucial step towards more specific qMRI interpretation.

We provide evidence from lipids samples and post-mortem data for the sensitivity of the MDM signatures to the molecular environment (Figs. 1e, 3b, and 5h). The variability of MDM values between human brain regions also correlated with specific gene-expression profiles (Fig. 4). While the comparison of in vivo human brain measurements to previously published ex vivo findings is based on two different datasets, these measurements are highly stable across normal subjects and the intersubject variabilities are much smaller than the regional variability. The agreement between the modalities provides strong evidence for the ability of our method to capture molecular information.

Remarkably, we were able to demonstrate the sensitivity of MDM signatures to lipid composition using direct comparison on post-mortem porcine brains. Even though there are many challenges in scanning post-mortem tissue, segmenting it, and comparing it to anatomically relevant histological results, we were able to replicate our in vivo findings. We provide histological validation for the MRI estimation of MTV. Moreover, we find that while standard qMRI parameters and MTV do not explain the lipidomic variability across the brain, the MDM signatures are in agreement with histological results. Lipids constitute the majority of the brain's dry weight and are known to be important for maintaining neural conduction and chemical balance[53,54]. The brain lipidome was shown to have a great deal of structural and functional diversity and was found to vary according to age, gender, brain region, and cell type[55]. Disruptions of the brain lipid metabolism have been linked to different disorders, including Alzheimer's disease, Parkinson's disease, depression, and anxiety[7,8,11,54–57]. Our results indicate that the MDM approach enhances the consistency between MRI-driven measurements and lipidomics, compared with standard qMRI parameters.

The simplicity of our model, which is based on a first-order approximation of qMRI dependencies, has great advantages in the modeling of complex environments. Importantly, we used lipids samples to show that the contributions of different mixture-components can be summed linearly (Fig. 1d). For contrast agents, the relaxivity is used to characterize the efficiency of different agents. Here, we treated the tissue itself, rather than a contrast material, as an agent to compute the relaxivity of the tissue. While relaxivity is usually calculated for R1 and R2, we extended this concept to other qMRI parameters. Our results showed that the tissue relaxivity changes as a function of the molecular composition. This suggests that the relaxivity of the tissue relates to the surface interaction between the water and the chemophysical environment. A theoretical formulation for the effect of the surface interaction on proton relaxation has been proposed before[58,59]. Specifically, a biophysical model for the linear relationship between R1 and R2 to the inverse of the water content $(1/WC = 1/(1 - MTV))$ was suggested by Fullerton et al.[43]. Interestingly, $1/WC$ varies almost linearly with MTV in the physiological range of MTV values. Applying our approach with $1/WC$ instead of MTV produces relatively similar results (Supplementary Fig. 28). However, using MTV as a measure of tissue relaxivity allowed us to generalize the linear model to multiple qMRI parameters, thus producing multidimensional MDM signatures.

We show that the MDM signatures allow for better understanding of the biological sources for the aging-related changes observe with MRI. Normal brain aging involves multiple changes, at both the microscale and macroscale levels. Measurements of macroscale brain volume have been widely used to characterize aging-associated atrophy. Our method of analysis can complement such findings and provide a deeper understanding of

microscale processes co-occurring with atrophy. Moreover, it allows us to test whether these various microscale and macroscale processes are caused by a common factor or represent the aging mosaic. Notably, we discovered that different brain regions undergo different biological aging processes. Therefore, combining several measurements of brain tissue is crucial in order to fully describe the state of the aged brain. For example, the macroscale aging-related volume reduction in cortical gray areas was accompanied by conserved tissue density, as estimated by MTV, and region-specific chemophysical changes, as estimated by the MDM. In contrast, in white-matter areas both MDM and MTV changed with age. These microscale alterations were not accompanied by macroscale volume reduction. Our in vivo results were validated by previous histological studies, which reported that the cortex shrinks with age, while the neural density remains relatively constant[49,50]. In contrast, white matter was found to undergo significant loss of myelinated nerve fibers during aging[51]. In addition, we found that the shrinkage of the hippocampus with age is accompanied with conserved tissue density and chemophysical composition. This is in agreement with histological findings, which predict drastic changes in hippocampal tissue composition in neurological diseases such as Alzheimer, but not in normal aging[49,50,60,61]. In contrast, hippocampal macroscale volume reduction was observed in both normal and pathological aging[2].

It should be noted that most of the human subjects recruited for this study were from the academic community. However, the different age groups were not matched for variables such as IQ and socioeconomic status. In addition, the sample size in our study was quite small. Therefore, the comparison we made between the two age groups may be affected by variables other than age. Our approach may benefit from validation based on larger quantitative MRI datasets[27,62]. Yet, we believe we have demonstrated the potential of our method to reveal molecular alterations in the brain. Moreover, the agreement of our findings with previous histological aging studies supports the association between the group differences we measured and brain aging. Our results suggest that the MDM approach may be very useful in differentiating the effects of normal aging from those of neurodegenerative diseases. There is also great potential for applications in other brain research fields besides aging. For example, our approach may be used to advance the study and diagnosis of brain cancer, in which the lipidomic environment undergoes considerable changes[63–65].

To conclude, we have presented here a quantitative MRI approach that decodes the molecular composition of the aging brain. While common MRI measurements are primarily affected by the water content of the tissue, our method employed the tissue relaxivity to expose the sensitivity of MRI to the molecular microenvironment. We presented evidence from lipid samples, post-mortem porcine brains and in vivo human brains for the sensitivity of the tissue relaxivity to molecular composition. Results obtained by this method in vivo disentangled different biological processes occurring in the human brain during aging. We identified region-specific patterns of microscale aging-related changes that are associated with the molecular composition of the human brain. Moreover, we showed that, in agreement with the mosaic theory of aging, different biological age-related processes measured in vivo have unique spatial patterns throughout the brain. The ability to identify and localize different age-derived processes in vivo may further advance human brain research.

## Methods

**Phantom construction**. The full protocol of lipids phantom preparation is described in Shtangel et al.[66].

In short, we prepared liposomes from one of the following lipids: phosphatidylserine (PS), phosphatidylcholine (PtdCho), phosphatidylcholine-cholesterol (PtdCho-Chol), Phosphatidylinositol-phosphatidylcholine (PI-PtdCho), or sphingomyelin (Spg). These phantoms were designed to model biological membranes and were prepared from lipids by the hydration–dehydration dry film technique[67]. The lipids were dissolved over a hot plate and vortexed. Next, the solvent was removed to create a dry film by vacuum-rotational evaporation. The samples were then stirred on a hot plate at 65 °C for 2.5 h to allow the lipids to achieve their final conformation as liposomes. Liposomes were diluted with Dulbecco's phosphate buffered saline (PBS), without calcium and magnesium (Biological Industries), to maintain physiological conditions in terms of osmolarity, ion concentrations and pH. To change the MTV of the liposome samples we varied the PBS to lipid volume ratios[66]. Samples were then transferred to the phantom box for scanning in a 4 mL squared polystyrene cuvettes glued to a polystyrene box, which was then filled with ~1% SeaKem Agarose (Ornat Biochemical) and ~0.0005 M Gd (Gadotetrate Melumine, (Dotarem, Guerbet)) dissolved in double distilled water (ddw). The purpose of the agar with Gd (Agar-Gd) was to stabilize the cuvettes, and to create a smooth area in the space surrounding the cuvettes that minimalized air–cuvette interfaces. In some of our experiments we used lipid mixtures composed of several lipids. We prepared nine mixtures containing different combinations of two out of three lipids (PtdChol, Spg and PS) in varying volume ratios (1:1,1:2,2:1). For each mixture, we prepared samples in which the ratio between the different lipid components remained constant while the water-to-lipid volume fraction varied.

For the bovine serum albumin (BSA) phantoms, samples were prepared by dissolving lyophilized BSA powder (Sigma Aldrich) in PBS. To change the MTV of these phantoms, we changed the BSA concentration. For the BSA + Iron phantoms, BSA was additionally mixed with a fixed concentration of 50 μg/mL ferrous sulfate heptahydrate (FeSO4*7H2O). Samples were prepared in their designated concentrations at room temperature. Prepared samples were allowed to sit overnight at 4 °C to ensure BSA had fully dissolved, without the need for significant agitation, which is known to cause protein cross-linking. Samples were then transferred to the phantom box for scanning.

For Glucose and Sucrose phantoms, different concentrations of D-( + )-Sucrose (Bio-Lab) and D-( + )-Glucose (Sigma) were dissolved in PBS at 40 °C. Samples were allowed to reach room temperature before the scan.

**MRI acquisition for phantoms**. Data was collected on a 3 T Siemens MAGNETOM Skyra scanner equipped with a 32-channel head receive-only coil at the ELSC neuroimaging unit at the Hebrew University.

For quantitative R1 & MTV mapping, three-dimensional (3D) Spoiled gradient (SPGR) echo images were acquired with different flip angles (α = 4°, 8°, 16°, and 30°). The TE/TR was 3.91/18 ms. The scan resolution was 1.1 × 1.1 × 0.9 mm. The same sequence was repeated with a higher resolution of 0.6 × 0.6 × 0.5 mm. The TE/TR was 4.45/18 ms. For calibration, we acquired an additional spin-echo inversion recovery (SEIR) scan. This scan was done on a single slice, with adiabatic inversion pulse and inversion times of TI = 2000, 1200, 800, 400, and 50. The TE/TR was 73/2540 ms. The scan resolution was 1.2 mm isotropic.

For quantitative T2 mapping, images were acquired with a multi spin-echo sequence with 15 equally spaced spin echoes between 10.5 ms and 157.5 ms. The TR was 4.94 s. The scan resolution was 1.2 mm isotropic. For quantitative MTsat mapping, images were acquired with the FLASH Siemens WIP 805 sequence. The TR was 23 ms for all samples except PI:PtdCho for which the TR was 72 ms. Six echoes were equally spaced between 1.93 ms to 14.58 ms. The on-resonance flip angle was 6°, the MT flip angle was 220°, and the RF offset was 700. We used 1.1-mm in-plane resolution with a slice thickness of 0.9 mm. For samples of sucrose and glucose, MTsat mapping was done similar to the human subjects, based on 3D Spoiled gradient (SPGR) echo image with an additional MT pulse. The flip angle was 10°, the TE/TR was 3.91/28 ms. The scan resolution was 1 mm isotropic.

**Estimation of qMRI parameters for phantoms**. MTV and R1 estimations for the lipids samples were computed based on a the mrQ[39] (https://github.com/mezera/mrQ) and Vista Lab (https://github.com/vistalab/vistasoft/wiki) software. The mrQ software was modified to suit the phantom system[66]. The modification utilizes the fact that the Agar-Gd filling the box around the samples is homogeneous and can, therefore, be assumed to have a constant T1 value. We used this gold standard T1 value generated from the SEIR scan to correct for the excite bias in the spoiled gradient echo scans. While the data was acquired in two different resolutions (see "MRI acquisition"), in our analysis we use the median R1 and MTV of each lipid sample and these are invariant to the resolution of acquisition (Supplementary Fig. 1e). Thus, we were able to use scans with different resolutions without damaging our results. T2 maps were computed by implementing the echo-modulation curve (EMC) algorithm[68].

For quantitative MTsat mapping see the "MTsat estimation" section for human subjects.

**MDM computation for phantoms**. We computed the dependency of each qMRI parameter (R1, MTsat, and R2) on MTV in different lipids samples. This process was implemented in MATLAB (MathWorks, Natwick, MI, USA). To manipulate the MTV values, we scanned samples of the same lipid in varying concentrations.

We computed the median MTV of each sample, along with the median of qMRI parameters. We used these data points to fit a linear model across all samples of the same lipid. The slope of this linear model represents the MTV derivative of the linear equation. We used this derivative estimate of three qMRI parameters (R1, R2, and MTsat) to compute the MDM signatures. The same procedure was used for the MDM computation of lipid mixtures.

**MDM modeling of lipid mixtures**. We tested the ability of MDM to predict the composition of lipid mixtures. For this analysis we used nine mixture phantoms (see "Phantom construction"), along with the three phantoms of the pure lipid constituents of the mixtures (PS, Spg, and Ptd-Cho).

In order to predict the qMRI parameters of a lipid mixture (Fig. 1d) we used Supplementary Eq. 1 (Supplementary Note 1). To further predict the composition of the mixtures (Fig. 1e) we used Supplementary Eq. 5 (Supplementary Note 2). We solved this equation using the QR factorization algorithm.

**Ethics**. Human experiments complied with all relevant ethical regations. The Helsinki Ethics Committee of Hadassah Hospital, Jerusalem, Israel approved the experimental procedure. Written informed consent was obtained from each participant prior to the procedure.

**Human subjects**. Human measurements were performed on 23 young adults (aged $27 \pm 2$ years, 11 females), and 18 older adults (aged $67 \pm 6$ years, five females). Healthy volunteers were recruited from the community surrounding the Hebrew University of Jerusalem.

**MRI acquisition for human subjects**. Data was collected on a 3 T Siemens MAGNETOM Skyra scanner equipped with a 32-channel head receive-only coil at the ELSC neuroimaging unit at the Hebrew University.

For quantitative R1, R2*, & MTV mapping, 3D Spoiled gradient (SPGR) echo images were acquired with different flip angles ($\alpha = 4°$, $10°$, $20°$, and $30°$). Each image included five equally spaced echoes (TE = 3.34–14.02 ms) and the TR was 19 ms (except for six young subjects for which the scan included only one TE = 3.34 ms). The scan resolution was 1 mm isotropic. For calibration, we acquired additional spin-echo inversion recovery scan with an echo-planar imaging (EPI) read-out (SEIR-epi). This scan was done with a slab-inversion pulse and spatial-spectral fat suppression. For SEIR-epi, the TE/TR was 49/2920 ms. TI were 200, 400, 1,200, and 2400 ms. We used 2-mm in-plane resolution with a slice thickness of 3 mm. The EPI read-out was performed using $2 \times$ acceleration.

For quantitative T2 mapping, multi-SE images were acquired with ten equally spaced spin echoes between 12 ms and 120 ms. The TR was 4.21 s. The scan resolution was 2 mm isotropic. T2 scans of four subjects (one young, three old) were excluded from the analysis due to motion.

For quantitative MTsat mapping, 3D Spoiled gradient (SPGR) echo image were acquired with an additional MT pulse. The flip angle was 10°, the TE/TR was 3.34/27 ms. The scan resolution was 1 mm isotropic.

Whole-brain DTI measurements were performed using a diffusion-weighted spin-echo EPI sequence with isotropic 1.5-mm resolution. Diffusion weighting gradients were applied at 64 directions and the strength of the diffusion weighting was set to $b = 2000$ s/mm$^2$ (TE/TR = 95.80/6000 ms, G = 45mT/m, $\delta = 32.25$ ms, $\Delta = 52.02$ ms). The data includes eight non-diffusion-weighted images ($b = 0$). In addition, we collected non-diffusion-weighted images with reversed phase-encode blips. For five subjects (four young, one old) we failed to acquire this correction data and they were excluded from the diffusion analysis.

Anatomical images were acquired with 3D magnetization prepared rapid gradient echo (MP-RAGE) scans for 24 of the subjects (14 from the younger subjects, 10 from the older subjects). The scan resolution was 1 mm isotropic, the TE/TR was 2.98/2300 ms. Magnetization Prepared 2 Rapid Acquisition Gradient Echoes (MP2RAGE) scans were acquired for the rest of the subjects. The scan resolution was 1 mm isotropic, the TE/TR was 2.98/5000 ms.

**Estimation of qMRI parameters for human subjects**. Whole-brain MTV and R1 maps, together with bias correction maps of B1 + and B1-, were computed using the mrQ software[39,69] (https://github.com/mezera/mrQ). Voxels in which the B1 + inhomogeneities were extrapolated and not interpolated were removed from the MTV and R1 maps. While we did not correct our MTV estimates for R2*, we showed that employing such a correction does not significantly change our results (see Supplementary Note 6, Supplementary Figs. 20–27). MTV maps of four subjects had bias in the lower part of the brain and they were therefore excluded from the analysis presented in Fig. 3, which includes ROIs in the brainstem.

Whole-brain T2 maps were computed by implementing the echo-modulation curve (EMC) algorithm[68]. To combine the MTV and T2 we co-registered the quantitative MTV map to the T2 map. We used the ANTS software package[70] to calculate the transformation and to warp the MTV map and the segmentation. The registration was computed to match the T1 map to the T2 map. Next, we applied the calculated transformation to MTV map (since MTV and T1 are in the same imaging space) and resampled the MTV map to match the resolution of the T2 map. The same transformation was also applied to the segmentation. R2 maps were calculated as 1/T2.

Whole-brain MTsat maps were computed as described in Helms et al.[37]. The MTsat measurement was extracted from Eq. (1):

$$\text{MTsat} = M_0 B1\alpha \frac{R1\text{TR}}{S_{\text{MT}}} - \frac{(B1\alpha)^2}{2} - R1\text{TR} \qquad (1)$$

Where $S_{\text{MT}}$ is the signal of the SPGR scan with additional MT pulse, $\alpha$ is the flip angle and TR is the repetition time. $M_o$ (the equilibrium magnetization parameter), B1 (the transmit inhomogeneity), and R1 estimations were computed from the non-MT weighted SPGR scans, during the pipeline described under "MTV & R1 estimation". Registration of the $S_{\text{MT}}$ image to the imaging space of the MTV map was done using a rigid-body alignment (R1, B1, and $M_O$ are all in the same space as MTV).

Diffusion analysis was done using the FDT toolbox in FSL[71,72]. Susceptibility and eddy current induced distortions were corrected using the reverse phase-encode data, with the eddy and topup commands[73,74]. MD maps were calculated using vistasoft (https://github.com/vistalab/vistasoft/wiki). We used a rigid-body alignment to register the corrected dMRI data to the imaging space of the MTV map (Flirt, FSL). In order to calculate the MD-MTV derivatives, we resampled the MTV map and the segmentation to match the dMRI resolution.

We used the SPGR scans with multiple echoes to estimate R2*. Fitting was done through the MPM toolbox[75]. As we had four SPGR scans with variable flip angles, we averaged the R2* maps acquired from each of these scans for increased SNR.

**Human brain segmentation**. Whole-brain segmentation was computed automatically using the FreeSurfer segmentation algorithm[76]. For subjects who had an MP-RAGE scan, we used it as a reference. For the other subjects the MP2RAGE scan was used as a reference. These anatomical images were registered to the MTV space prior to the segmentation process, using a rigid-body alignment. Sub-cortical gray-matter structures were segmented with FSL's FIRST tool[77]. To avoid partial volume effects, we removed the outer shell of each ROI and left only the core.

**MDM computation in the human brain**. We computed the dependency of each qMRI parameter (R1, MTsat, MD, and R2) on MTV in different brain areas. This process was implemented in MATLAB (MathWorks, Natwick, MI, USA). For each ROI, we extracted the MTV values from all voxels and pooled them into 36 bins spaced equally between 0.05 and 0.40. This was done so that the linear fit would not be heavily affected by the density of the voxels in different MTV values. We removed any bins in which the number of voxels was smaller than 4% of the total voxel count in the ROI. The median MTV of each bin was computed, along with the median of the qMRI parameter. We used these data points to fit the linear model across bins using Eq. (2):

$$\text{qMRI parameters} = a * \text{MTV} + b \qquad (2)$$

The slope of this linear model ("$a$") represents the MTV derivative of the linear equation. We used this derivative estimate to compute the MDM signatures.

For each subject, ROIs in which the total voxel count was smaller than a set threshold of 500 voxels for the MTsat and R1 maps, 150 voxels for the MD map, and 50 voxels for the R2 map were excluded.

**Principal component analysis (PCA) in the human brain**. To estimate the variability in the MDM signatures across the brain, we computed the first principal component (PC) of MDM. For each MDM dimension (MTV derivatives of R1, MTsat, MD, and R2), we evaluated the median of the different brain areas across the young subjects. As each MDM dimension has different units, we then computed the $z$-score of each dimension across the different brain area. Finally, we performed PCA. The variables in this analysis were the different MDM dimensions, and the observations were the different brain areas. From this analysis, we derived the first PC that accounts for most of the variability in MDM signatures across the brain. To estimate the median absolute deviations (MAD) across subjects of each MDM measurement in the PC basis, we applied the $z$-score transformation to the original MAD and then projected them onto the PC basis.

To compute the first PC of standard qMRI parameters we followed the same procedure, but used R1, MTsat, MD, and R2 instead of their MTV derivatives.

For the first PC of molecular composition, we followed the same procedure, but used the phospholipid composition and the ratio between phospholipids to proteins and cholesterol as variables. The data was taken from eight post-mortem human brains[7]. Brains were obtained from individuals between 54 and 57 years of age, which were autopsied within 24 h after death.

**Linear model for prediction of human molecular composition**. We used MDM measurements in order to predict the molecular composition of different brain areas (Fig. 3c). For this analysis we used Supplementary Eq. 5 in the Supplementary Note 2. We solved this equation using QR factorization algorithm (for more details see Supplementary Note 3).

**Gene-expression dataset**. For the gene-expression analysis we followed the work of Ben-David and Shifman[46]. Microarray data was acquired from the Allen Brain Atlas (http://human.brain-map.org/well_data_files) and included a total of 1340 microarray

profiles from donors H0351.2001 and H0351.2002, encompassing the different regions of the human brain. The donors were 24 and 39 years old, respectively, at the time of their death, with no known psychopathologies. We used the statistical analysis described by Ben-David and Shifman[46]. They constructed a gene network using a weighted gene co-expression network analysis. The gene network included 19 modules of varying sizes, from 38 to 7385 genes. The module eigengenes were derived by taking the first PC of the expression values in each module. In addition, we used the gene ontology enrichment analysis described by Ben-David and Shifman to define the name of each module. The colors of the different modules in the Fig. 4 and Supplementary Fig. 10 are the same as in the original paper.

Next, we matched between the gene-expression data and the MRI measurements. This analysis was done on 35 cortical regions extracted from FreeSurfer cortical parcellation. We downloaded the T1-weighted images of the two donors provided by the Allen Brain Atlas (http://human.brain-map.org/mri_viewers/data) and used them as a reference for FreeSurfer segmentation. We then found the FreeSurfer label of each gene-expression sample using the sample's coordinates in brain space. We removed samples for which the FreeSurfer label and the label provided in the microarray dataset did not agree (there were 72 such samples out of 697 cortical samples). For each gene module, we averaged over the eigengenes of all samples from the same cortical area across the two donors.

Last, we compared the cortical eigengene of each module to the projection of cortical areas on the first PC of MDM. In addition, we compared the modules' eigengenes to the MTV values of the cortical areas and to the projection of cortical areas on the first PC of standard qMRI parameters (Supplementary Fig. 10). These 57 correlations were corrected for multiple comparisons using the FDR method.

**Brain region's volume computation**. To estimate the volume of different brain regions, we calculated the number of voxels in the FreeSurfer segmentation of each region (see "Brain segmentation").

**R2\* correction for MTV**. To correct the MTV estimates for R2* we used Eq. (3):

$$MTV_C = 1 - (1 - MTV) \cdot \exp(TE \cdot R2^*) \tag{3}$$

Where $MTV_C$ is the corrected MTV.

**Statistical analysis**. The statistical significance of the differences between the age groups was computed using an independent-sample $t$-test (alpha = 0.05, both right and left tail) and was corrected for multiple comparisons using the false-discovery rate (FDR) method. For this analysis, MRI measurements of both hemispheres of bilateral brain regions were joined together. $R^2$ measurements were adjusted for the number of data points. All statistical tests were two-sided.

**Post-mortem tissue acquisition**. Two post-mortem porcine brains were purchased from BIOTECH FARM.

**Post-mortem MRI acquisition**. Brains were scanned fresh (without fixation) in water within 6 h after death. Data was collected on a 3 T Siemens MAGNETOM Skyra scanner equipped with a 32-channel head receive-only coil at the ELSC neuroimaging unit at the Hebrew University.

For quantitative R1, R2*, & MTV mapping, 3D Spoiled gradient (SPGR) echo images were acquired with different flip angles ($\alpha$ = 4°, 10°, 20°, and 30°). Each image included five equally spaced echoes (TE = 4.01 – 16.51 ms) and the TR was 22 ms. The scan resolution was 0.8 mm isotropic. For calibration, we acquired additional spin-echo inversion recovery scan with an echo-planar imaging (EPI) read-out (SEIR-epi). This scan was done with a slab-inversion pulse and spatial-spectral fat suppression. For SEIR-epi, the TE/TR was 49/2920 ms. TI were 50, 200, 400, 1200 ms. The scan resolution was 2 mm isotropic. The EPI read-out was performed using 2 × acceleration.

For quantitative T2 mapping, multi-SE images were acquired with ten equally spaced spin echoes between 12 and 120 ms. The TR was 4.21 s. The scan resolution was 2 mm isotropic.

For quantitative MTsat mapping, 3D Spoiled gradient (SPGR) echo image were acquired with an additional MT pulse. The flip angle was 10°, the TE/TR was 4.01/40 ms. The scan resolution was 0.8 mm isotropic.

Whole-brain DTI measurements were performed using a diffusion-weighted spin-echo EPI sequence with isotropic 1.5-mm resolution. Diffusion weighting gradients were applied at 64 directions and the strength of the diffusion weighting was set to $b$ = 2000 s/mm$^2$ (TE/TR = 95.80/6000 ms, $G$ = 45mT/m, $\delta$ = 32.25 ms, $\Delta$ = 52.02 ms). The data includes eight non-diffusion-weighted images ($b$ = 0).

For anatomical images, 3D magnetization prepared rapid gradient echo (MP-RAGE) scans were acquired. The scan resolution was 1 mm isotropic, the TE/TR was 2.98/2300 ms.

**Histological analysis**. Following the MRI scans the brains were dissected. Total of 42 brain regions were identified. Four samples were excluded as we were not able to properly separate the WM from the GM. One sample was excluded as we could not properly identify its anatomical origin. Additional two samples were too small for TLC analysis.

The non-water fraction (MTV) was determined by desiccation, also known as the dry-wet method. A small fraction of each brain sample (~0.25 g) was weighed. In order to completely dehydrate the fresh tissues, they were left for several days in a vacuum dessicator over silica gel at 4 °C. The experiment ended when no further weight loss occurred. The MTV of each brain sample was calculated based on the difference between the wet ($W_{wet}$) and dry ($W_{dry}$) weights of the tissue (Eq. 4):

$$MTV = \frac{W_{wet} - W_{dry}}{W_{wet}} \tag{4}$$

For lipid extraction and lipidomics analysis[78], Brain samples were weighted and homogenized with saline in plastic tubes on ice at concentration of 1 mg/12.5 µL. Two-hundred fifty microliters from each homogenate were utilized for lipid extraction and analysis with thin-layer chromatography (TLC). The lipid species distribution was analyzed by TLC applying 150 µg aliquots. Samples were reconstituted in 10 µL of Folch mixture and spotted on Silica-G TLC plates. Standards for each fraction were purchased from Sigma Aldrich (Rehovot, Israel) and were spotted in separate TLC lanes, i.e., 50 µg of triacylglycerides (TG), cholesterol (Chol), cholesteryl esters (CE), free fatty acids (FFA), lysophospholipids (Lyso), sphingomyelin (Spg), phosphatidylcholine (PtdCho), phosphatidylinositol (PI), phosphatidylserine (PS), and phosphatidylethanolamine (PE). Plates were then placed in a 20 × 20 cm TLC chamber containing petroleum ether, ethyl ether, and acetic acid (80:20:1, $v/v/v$) for quantification of neutral lipids or chloroform, methanol, acetic acid, and water (65:25:4:2, $v:v:v:v$) for quantification of polar lipids and run for 45 min. TG, Chol, CE, FFA, phospholipids (PL), Lyso, Spg, PtdCho, PI, PS, and PE bands were visualized with Iodine, scanned and quantified by Optiquant after scanning (Epson V700). Lyso, CE, TG, and PI were excluded from further analysis as their quantification was noisy and demonstrated high variability across TLC plates. This analysis was conducted under the guidance of Prof. Alicia Leikin-Frenkel in the Bert Strassburger Lipid Center, Sheba, Tel Hashomer.

**Estimation of qMRI parameters in the post-mortem brain**. Similar to human subjects.

**Brain segmentation of post-mortem brain**. Brain segmentation was done manually. Five tissue samples were excluded as we could not identify their origin location in the MRI scans.

**MDM computation in the post-mortem brain**. We computed the dependency of each qMRI parameter (R1, MTsat, MD, and R2) on MTV in different brain areas similarly to the analysis of the human subjects.

**Principal component analysis (PCA) in the post-mortem brain**. To estimate the variability in the MDM signatures across the brain, we computed the first principal component (PC) of MDM. PCA analysis was performed with four variables corresponding to the MDM dimensions (MTV derivatives of R1, MTsat, MD, and R2), and 30 observations corresponding to the different brain regions. As each MDM dimension has different units, we first computed the $z$-score of each dimension across the different brain areas prior to the PCA. From this analysis we derived the first PC that accounts for most of the variability in MDM signatures across the brain.

To compute the first PC of standard qMRI parameters we followed the same procedure, but used R1, MTsat, MD, and R2 instead of their MTV derivatives.

To estimate the variability in the lipid composition across the brain, we computed the first principal component (PC) of lipidomics. PCA analysis was performed with seven variables corresponding to the different polar and neutral lipids (Chol, FFA, PL, Spg, PtdCho, PS, PE), and 30 observations corresponding to the different brain regions. From this analysis, we derived the first PC that accounts for most of the variability in lipid composition across the brain.

**Reporting summary**. Further information on research design is available in the Nature Research Reporting Summary linked to this article.

## Data availability
The datasets generated and/or analyzed during the current study are available from the corresponding author on reasonable request.

## Code availability
A toolbox for computing MDM signatures is available at [https://github.com/MezerLab/MDM_toolbox].

The code generating the figures of in the paper is available at [https://github.com/MezerLab/MDM_Gen_Figs].

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

## Acknowledgements

This work was supported by the ISF grant 0399306, awarded to A.A.M. We acknowledge Ady Zelman for the assistance in collecting the human MRI data. We thank Assaf Friedler for assigning research lab space and advising on the lipid sample experiments. We thank Inbal Goshen for assigning research lab space and advising on the protein and ion samples as well as the porcine brain experiments. We thank Magnus Soderberg for advising on histological data interpretation. We are grateful to Brian A. Wandell, Jason Yeatman, Hermona Soreq, Ami Citri, Mark Does, Yaniv Ziv, Ofer Yizhar, Shai Berman, Roey Schurr, Jonathan Bain, Asier Erramuzpe Aliaga, Menachem Gutman, and Esther Nachliel for their critical reading of the manuscript and very useful comments. We thank Prof. Alicia Leikin-Frenkel for her guidance with the TLC analysis. We thank Rona Shaharabani for guidance and support in the post-mortem experiments.

## Author contributions

S.F., O.S., and A.A.M. conceived of the presented idea. S.F. and A.A.M. wrote the manuscript and designed the figures. S.F. collected the human and non-human brain datasets and analyzed them. O.S. performed the phantom experiments and analyzed them. B.W. performed the phantom experiments for non-lipid compounds. N.S. performed the gene-expression analysis. S.S. assisted and instructed with the gene-expression analysis. A.K. performed the porcine brain dissection.

## Additional information

**Competing interests:** A.A.M, S.F., O.S. and the Hebrew University of Jerusalem have filed a patent application describing the technology used to measure MDM in this work. The other authors declare no competing interests.

