## [Peer Review File · Nature Communications]

Reviewers' comments:

Reviewer #1 (Remarks to the Author):

This is an interesting advance in MRI-based methods for evaluating the molecular composition of brain tissue in living humans subjects.

However, in the end, it is not clear exactly what molecules they are measuring and how it relates to brain function or disease. Their claim that this is a major advance is unjustified. Certainly measuring specific neurochemicals in living subjects is equally or even more valuable.

Besides water and lipids, the brain contains numerous small molecules, from ions to metabolites, and other macromolecules including proteins and sugars. All of these chemicals are conspicuously ignored in this paper. Are we to believe that these other molecules play no role in influencing inherent tissue relaxivity and generating the particular qMR measures introduced here?

While a phantom has to be simpler than the actual tissue it models, it should not be too simple. For instance, they could have created phantoms of variable lipid composition in terms of liposomes ("designed to model cellular membranes") but recreated the known ionic composition of the brain's interstitial fluid and comparable protein content.

The validation process for this method proceeds from phantom models of known composition, to comparison with historical data on brain composition, to testing of human subjects. Could they have scanned frozen brain tissue for a direct comparison with its chemical composition? The title of "Figure 3: The biological interpretation of the MDM signatures based on post mortem data" is somewhat misleading in that regard.

There are some inconsistencies in the statistical methods used in every step in the validation. For instance, Fig 3 makes use of the "1st PC of in-vivo MDM on the cortical surface". Why don't they show its relation to specific lipids. Moreover, if this is an effective measure for reducing the data complexity, why do the authors no longer use it in their testing of human subjects, such as in Figs. 4 and 5? On the other hand, why did they not examine the relationship between MDM measures and individual lipid composition.

Reviewer #2 (Remarks to the Author):

This is a very interesting paper, proposing a refreshing approach to multi-parametric quantitative MRI. The Authors proposed to investigate, instead of a combination of many MRI parameters, the dependency of specific parameters on non-water content (i.e., the molecular composition of tissue). Using phantoms, they demonstrate that the dependency of qMRI parameters on non-water content is linear, and can be used to extrapolate a unique signature for specific lipids. They then go onto demonstrating that these dependency vary across brain regions, and that this variance reflects the region-specific molecular composition measured ex-vivo. They also show some association with gene-expression in membranes. Finally, they use this method to address the question whether brain changes occurring in aging are driven by a common cause or by a "mosaic" of con-causes.

The paper is elegant and well written. The figures are of high quality, and the amount of work that went into this manuscript is substantial.

Comments/Suggestions:

1) My main criticism is that the results are obtained through a fair amount of data manipulation, which makes it a bit difficult to follow every step and spot potential circularity in the methods. In other words, I am wondering whether any of the results could be simply explained by the fact that the same MRI measurements are used many times.

2) Another concern that I have (which is probably due to my limited understanding) is that for some (possibly all) of the qMRI parameters used in the manuscript their interdependence is known or at least presumed. Particularly the dependency on water content (proton density). And since the macromolecular content is given by $(1-wc)$, it should be possible to compute the analytical derivatives of the given MRI parameters with respect to non water content to predict their trend. I would then expect the data presented in this manuscript to agree with this predicted trend. The simplest example is R1, for which a relationship of the form $R1 = a/PD + b$ is typically assumed (and shown in a previous paper of the senior author of this manuscript).

Based on this assumption, and on the knowledge that $MTV = \sim(1-PD)$, I struggle to understand the linear relationship between R1 and MTV shown in this paper. It would be useful if the Authors could explain how their data driven assessments relate to the expected behaviour.

3) The correlation between MDM and gene expression patterns (Fig 3E) is not very convincing, as also shown by the R^2 of 0.3. I am not suggesting to remove it completely but the Authors should mitigate their claims relative to this particular result, which could be driven by few outliers.

Minor comment:

The TR of the MTsat acquisition in humans seems very long - check this is the correct value.

Reviewer #3 (Remarks to the Author):

The authors used 3 quantitative MRI parameters (R1, R2, MTSat) to estimate lipid and water content of the brain and relate this to changes attributed with age. They used phantoms with different lipid contents, literature values from PM studies for lipid content of different brain regions, literature data on some genes in different cortical regions, and data from 23 younger and 18 older humans. There is substantial interest in characterising brain molecular changes with ageing non-invasively and MRI is a major tool for this. However, there are problems with the present approach which limit the interpretation and reliability of the results.

There are other quantitative MRI techniques which show sensitive and specific age related changes - eg several DTI parameters, T1 mapping for brain water, as well as volume loss and lesion formation. The present paper hardly mentioned these and did not compare their multi-sequence method with other more researched approaches.

The human subjects are not described at all - were these normal volunteers? What were their medical histories? How were they recruited? Were the young and old groups matched for premorbid IQ which has a major effect on white matter and cortical parameters? Were they matched for socioeconomic status, which also affects brain structure. Other variables such as educational attainment are important and should be balanced between the groups. If the groups were not balanced for these key factors, then other sources of variation between the individuals could account for observed differences, not age. The lack of characterisation of the humans is a major shortcoming in a paper that purports to present variables that are makers of age.

The phantom data have limitations - while several different lipids and concentrations were tested, phantoms cannot represent the complexity of human tissue, including not just the concentration of particular molecules but also their binding state, etc.

Use of literature values for regional brain lipids are a serious limitation. Were the papers on human brain? Were there any details about the brains that provided the samples described in the literature? How many brains contributed and what ages were they? Why not study PM human brain tissue itself with MRI then histology? This would be much more relevant, so that the MRI signatures could be compared directly with the relevant specimen in a PM MRI-histology protocol were used. Several have been developed by other groups. This enables a direct tissue to tissue comparison to be done.

The genetics - there are thousands of genes in the human brain that have associations with a range of diseases of aging - the present analysis was barely described and made little contribution to the reliability of the proposed quantitative MRI markers.

Some of the background information was rather overstated. For example, there are many studies ageing in the brain, or diseases that increase with age or accelerate aging, that do not support the concept of 'MRI changes of aging that are usually attributed to myelin' - what is this? Myelin is one of the last components of the brain to change with ageing or indeed in several pathologies such as age related white matter hyperintensities. Advanced quantitative MRI clearly shows that these areas represent increased water content at the earliest stage and even chronically. Several studies show that increase in tissue water content is one of the earliest changes with age, which with other pathologies eventually leads to myelin damage, axon and myelin loss - the idea that myelin damage or loss is an early feature of ageing reflects a longstanding artefact of tissue pathology - namely that the tissue rarefaction is due to water increase not myelin loss, but the water is routinely removed in the histopathology preparation therefore unobserved.

Dear reviewers,

Thank you for your thoughtful comments, and for agreeing to review the manuscript entitled "Non-invasive detection of age-related molecular profiles in the human brain" (NCOMMS-18-32563-T).

Before individually responding to the different comments made on the manuscript, we would like to mention a few general points. We revised the manuscript and improved it according to the reviewers' suggestions. Mainly, we performed a new post-mortem analysis for direct comparison of MDM signatures and molecular composition in porcine brains. This additional analysis provides further validation for the sensitivity of the MDM approach to the chemophysical microenvironment and the lipids composition. Other conventional quantitative MRI measures did not display this molecular sensitivity. In addition, we performed a series of new phantom experiments demonstrating the sensitivity of our MRI framework to differences not only between lipids but also between sugars, proteins and ions. Finally, we slightly modified the process of T1 and MTV mapping for human subject (specifically we upgraded the B1 estimation) and therefore the relevant figures in the manuscript have been updated.

The manuscript has gone through considerable revisions, including changes to the text as well as one additional main figure, and several additional supplementary figures. Below, we describe the changes made to the manuscript following each comment. The comments are in blue text, our responses are in black, and the revised text is in gray.

Reviewer #1:

This is an interesting advance in MRI-based methods for evaluating the molecular composition of brain tissue in living humans subjects. However, in the end, it is not clear exactly what molecules they are measuring and how it relates to brain function or disease. Their claim that this is a major advance is unjustified. Certainly measuring specific neurochemicals in living subjects is equally or even more valuable.

We thank the reviewer for finding our work interesting. We would like to stress that there is a great effort in the MRI community to provide in-vivo histology. Yet, current state-of-the-art whole brain quantitative MRI techniques does not go beyond "myelin fraction" estimation (Does, NeuroImage 2018, Weiskopf, Current Opinion in Neurology 2015), which is heavily dependent on the water content. Our approach, on the contrary, provides for the first-time sensitivity to the molecular composition of myelin and other sub-cellular compartments, which was validated on phantoms and on post-mortem brains (fig. 3B and our new analysis in fig. 4H). This molecular information is not captured by other widespread quantitative MRI techniques (Sup. Fig. 9BC and Fig. 4D-E). Moreover, we show that our approach can predict the fraction of specific phospholipids both in phantoms and in the brain (fig. 1E & fig. 3C).

While we agree that evaluating neurochemicals will be very interesting, we feel that estimation of molecular composition would be highly relevant for understanding brain function and disease. Lipids make the majority of the human brain in dry weight, and have broad information carrying roles in the CNS. The cell membranes, which are mainly composed of lipids, are critical in maintaining chemical balance in the brain. An abnormal lipid composition is likely to influence important physiological processes associated with ion

channels or receptor functions. In recent years there is a growing interest in post-mortem brain lipidomics (Piomelli et. al., Nature Reviews Neuroscience 2007, Lauwers, Neuron 2016). The brain lipidome was found to have a great deal of structural and functional diversity. It varies according to age, gender, brain region and cell type. Moreover, lipidome changes are associated with a wide spectrum of neurological and psychiatric diseases including Parkinson's disease, Alzheimer's disease, depression and anxiety. There is a very serious need for in-vivo mapping of molecular variations in different human brain disorders in order to capture failures in regulation of biochemical pathways and networks. Our introduction section line 25-26 mentions this. **Thanks to the reviewer's comment we now highlight this point also in the discussion (lines 310-316).**

Of course, there are many other subcellular components that are interesting. However, existing water protons MRI measurements cannot detect any of them. The well-established field of MR spectroscopy aims at identifying neurochemicals in the brain, yet this approach has a known SNR limitations. We want to emphasize that currently, there is no in-vivo imaging method that was shown to measure specific microenvironment components with whole brain coverage and 1mm isotropic resolution. In that sense, our work opens the door to a specific characterization of brain tissue, that until now was only possible post-mortem. We therefore believe that our novel in-vivo measurements will be highly valuable in the diagnostics and research of brain function and disease.

Besides water and lipids, the brain contains numerous small molecules, from ions to metabolites, and other macromolecules including proteins and sugars. All of these chemicals are conspicuously ignored in this paper. Are we to believe that these other molecules play no role in influencing inherent tissue relaxivity and generating the particular qMR measures introduced here?

We thank the reviewer for this highly relevant comment. We agree with the reviewer that besides lipids and water, the tissue also contains ions, metabolites, proteins and sugars. In fact, we never intended to claim that our method is exclusively sensitive to lipids and we are sorry it wasn't clear in the manuscript.

Our approach introduces for the first time the concept of tissue reflexivity. This novel measurement provides in-vivo access to the different tissue constituents. In the MRI literature, there are growing evidence that lipids are the major source of MRI signal in the brain (Leuze, Neuroimage, 2017). Nevertheless, we agree with the reviewer that our approach may captures many different components of the molecular microenvironment. Already in the original submission, we showed that our method allows prediction of the ratio of phospholipids to proteins (Fig. 3), and in addition we evaluated the effect of iron on our findings (Sup. Fig. 21). **Following this constructive comment, we have performed several additional phantom experiments showing that sugars, proteins and ions can influence relaxivity and be distinguished using the MDM signatures (Sup. Fig. 1D, for text see lines 94-98).** We also highlight this point in the discussion (lines 275-280): *"We chose to validate our approach using membrane lipids based on previous experiments⁵⁸⁻⁶². Nevertheless, we acknowledge the fact that brain tissue is comprised of many other compounds beside lipids, such as proteins, sugars and ions. As we have shown, these other compounds also exhibit unique dependency*

on MTV, thus demonstrating the comprehensive nature of our framework. The effect of such compounds is probably captured when we apply the MDM approach to the in-vivo human brain".

While a phantom has to be simpler than the actual tissue it models, it should not be too simple. For instance, they could have created phantoms of variable lipid composition in terms of liposomes ("designed to model cellular membranes") but recreated the known ionic composition of the brain's interstitial fluid and comparable protein content.

We thank the reviewer for raising this point.

First, in all of our phantom experiments liposomes were diluted with Dulbecco's Phosphate Buffered Saline (PBS), without calcium and magnesium, to maintain physiological conditions in terms of osmolarity, ion concentrations and PH. We apologize it wasn't described in the text before, but it is now stated in the methods section.

Second, we are sorry it wasn't clear enough in the manuscript, but the goal of our phantoms was no to mimic the full complexity of brain tissue. For this aim we applied our method for the in-vivo human brain and compared it to postmortem data. Nevertheless, we performed the phantom experiments in order to verify that our relaxivity method captures information regarding the chemophysical environment. Particularly, the phantom experiments show examples in which standard qMRI parameters cannot separate between changes in water fraction and molecular composition (Fig 1A insert). We show this ambiguity can be resolved using the MDM approach (Fig 1C). It is the simplicity of the phantoms that allowed us to prove the sensitivity of our method to very specific molecular components. Using the phantom system, we were able to show the sensitivity of our method to different brain lipids. Moreover, by evaluating mixtures of several lipids, we showed that the composition of a mixture can be predicted using our method. Such accurate predictions of molecular environment with MRI were never shown before. Finally, we show that the same model of prediction can be applied for the human brain to generate accurate predictions of specific molecular features (Fig. 3C).

Following the reviewer's comment, we highlighted this point in the results section (lines 68-70,94-98): *"Using this (phantom) system, we tested whether accounting for the effect of the water content on qMRI parameters will provide sensitivity to fine molecular details such as the head groups that distinguish different membrane phospholipids... The lipids samples do not simulate the full complexity of biological tissues. However, they allow us to estimate the sensitivity of our MRI approach to subtle chemophysical differences between phospholipids, in a completely controlled environment. Moreover, our approach can be applied to other compounds to reveal differences in the MRI signal between different proteins, sugars and ions (Sup. Fig. 1D)"*

Further clarification was added to the discussion (lines 269-275,280-282): *"We developed a unique phantom system of lipid samples to validate our method.... Many previous publications assume that MRI measurements are sensitive to different cellular components such as myelin. Our work tests such assumptions on a completely controlled phantom environment in a rigorous manner. While the phantom system is clearly simplistic compared to brain tissue, its simplicity allowed us to verify the specificity of our method to the chemophysical environment.*

Remarkably, our approach revealed unique signatures for different lipids, and is therefore sensitive even to the relatively subtle details that distinguish one lipid from another... Therefore, the phantoms were made to examine the MRI sensitivity for the chemophysical environment, and the human brain data was used to measure the true biological effects in a complex in-vivo environment".

The validation process for this method proceeds from phantom models of known composition, to comparison with historical data on brain composition, to testing of human subjects. Could they have scanned frozen brain tissue for a direct comparison with its chemical composition? The title of "Figure 3: The biological interpretation of the MDM signatures based on post mortem data" is somewhat misleading in that regard.

We agree with the reviewer that while post-mortem analysis is very challenging there is great importance in performing histological analysis and MRI scans on the same brain. Following the reviewer's suggestion, **we made a direct comparison of MRI measurements and histological analysis on post-mortem brains.**

For this aim we performed MRI scans (R1, MTsat, R2, MD and MTV mapping) followed by thin-layer chromatography (TLC) technique of two fresh post-mortem porcine brains. To our knowledge this is the first post-mortem analysis that compares between lipidomics and MRI. Even though there are many challenges in scanning post-mortem tissue, segmenting it, and comparing it to anatomically relevant histological result, we were able to replicate our findings in the level of the single brain. First, MTV values estimated using MRI were in agreement with the non-water fraction found histologically (adjusted $R^2=0.64$, $p<0.001$). Moreover, while MTV and standard qMRI parameters could not explain the lipidomics variability in the brain, the MDM signatures correlated significantly with the lipid composition (adjusted $R^2=0.3$, $p<0.01$). Excluding 5 brain regions (out of 30) with extreme TLC values yielded even stronger correlation between MDM signatures and molecular composition (adjusted $R^2=0.55$, $p<0.001$). The full analysis has been incorporated in the manuscript (Fig. 4, and without outliers in Sup. Fig. 10C). We believe it greatly strengthens the credibility of our approach and demonstrates its specificity to brain lipids. In addition, we provide for the first-time histological validation for the MRI estimation of MTV on brain tissue.

However, we want to stress the validity of the comparison we made in the original submission based on histological data from the literature (Fig. 3). Even though our data was averaged over 8 human brains for the histological measurements and 23 human brains for the MRI signatures, the intersubject variabilities are much smaller than the regional variability. Note that in figure 3B the error bars are small relative to the variability between brain areas. Therefore, despite the fact that the human histological and in-vivo measurements were not taken from the same tissue, these measurements are highly stable across normal subjects and the astonishing agreement between the two modalities ($R^2=0.84$) provides a strong evidence for the ability of our method to capture molecular information. We would like to point out that using standard qMRI parameters instead of our new measurements yielded much lower agreement between histological and in-vivo or ex-vivo MRI measures (sup. figures 9B-C & 10C). Following the reviewer's comment, we adapted the discussion (lines 298-304).

We are sorry the title was misleading, we changed it to: "*The biological interpretation of the MDM signatures based on comparison between in-vivo and post mortem data*". In addition, we added a clarification in the figure caption: "*Comparison of the in-vivo MDM signatures of different brain regions to the molecular composition of these regions as reported in the literature for 8 post-mortem human brains (Söderberg, 1990)*".

There are some inconsistencies in the statistical methods used in every step in the validation. For instance, Fig 3 makes use of the "1st PC of in-vivo MDM on the cortical surface". Why don't they show its relation to specific lipids. Moreover, if this is an effective measure for reducing the data complexity, why do the authors no longer use it in their testing of human subjects, such as in Figs. 4 and 5? On the other hand, why did they not examine the relationship between MDM measures and individual lipid composition.

We thank the reviewer for this comment. Indeed, in figure 3 (and now also in the new figure 4) we use PCA to reduce the dimensionality of the MDM signatures. Originally, we did not examine the relationship between the 1st PC of MDM and individual lipid composition in order to avoid many comparisons (as there are seven different histological measures). **Following the reviewer's suggestion, we added this analysis to the manuscript (Sup. Fig. 9A).** Interestingly, the 1st PC of MDM correlates significantly with the fractions of phosphatidylethanolamine (PE), phosphatidylcholine (PtdCho), sphingomyelin (Spg) and the ratio between phospholipids and proteins.

While we believe that the 1st PC of MDM is an effective measure, the reviewer is right to point out that we no longer use it in the aging analysis (Fig. 5-6). In this analysis, we wanted to compare aging-related changes revealed by standard qMRI parameters to the separation of these qMRI parameters to molecular and water related contributions. Our goal was to demonstrate that in accordance with our hypothesis, changes in MRI measurements observed with aging result from a combination of alterations in the molecular composition of the tissue and its water content. **We now show also a comparison of the aging related changes in the 1st PC of MDM, the 1st PC of standard qMRI parameters and MTV (Sup. Fig. 18).** Interestingly, the 1st PC of MDM reveals aging related changes not captured by the 1st PC of standard qMRI parameters or MTV.

Reviewer #2:

This is a very interesting paper, proposing a refreshing approach to multi-parametric quantitative MRI. The Authors proposed to investigate, instead of a combination of many MRI parameters, the dependency of specific parameters on non-water content (i.e., the molecular composition of tissue). Using phantoms, they demonstrate that the dependency of qMRI parameters on non-water content is linear, and can be used to extrapolate a unique signature for specific lipids. They then go onto demonstrating that these dependency vary across brain regions, and that this variance reflects the region-specific molecular composition measured ex-vivo. They also show some association with gene-expression in membranes. Finally, they use this method to address the question whether brain changes occurring in aging are driven by a common cause or by a "mosaic" of con-causes.

The paper is elegant and well written. The figures are of high quality, and the amount of work that went into this manuscript is substantial.

We thank the reviewer for appreciating our work and acknowledging its novelty.

My main criticism is that the results are obtained through a fair amount of data manipulation, which makes it a bit difficult to follow every step and spot potential circularity in the methods. In other words, I am wondering whether any of the results could be simply explained by the fact that the same MRI measurements are used many times.

We thank the reviewer for raising this point. Following this concern, **we added an explanatory figure** of the different processing steps of the MDM approach (Sup. Fig. 7). We want to stress that we use several qMRI measurements to calculate the tissue's multidimensional MDM signatures. It is true that MTV is used when calculating the relaxivity of each qMRI parameter. However, the main point of this work is that many qMRI parameters are sensitive to the non-water fraction estimated directly by MTV (Fig. 1-2 & Sup. Fig. 3-5). Importantly we show that MTV by itself cannot be used to identify different lipids (Fig. 1A insert), is not correlated with molecular composition (Fig. 4D) and captures different aging effects relative to MDM (fig 5-7). Moreover, all the analyses are statistically corrected for multiple comparisons and the coefficient of determination used is adjusted for the effect of the number of data points.

Another concern that I have (which is probably due to my limited understanding) is that for some (possibly all) of the qMRI parameters used in the manuscript their interdependence is known or at least presumed. Particularly the dependency on water content (proton density). And since the macromolecular content is given by $(1-wc)$, it should be possible to compute the analytical derivatives of the given MRI parameters with respect to non water content to predict their trend. I would then expect the data presented in this manuscript to agree with this predicted trend. The simplest example is R1, for which a relationship of the form $R1=a/PD + b$ is typically assumed (and shown in a previous paper of the senior author of this manuscript).

Based on this assumption, and on the knowledge that $MTV = \sim(1-PD)$, I struggle to understand the linear relationship between R1 and MTV shown in this paper. It would be useful if the Authors could explain how their data driven assessments relate to the expected behaviour.

We thank the reviewer for this insightful comment. Indeed, a biophysical model for the linear relationship between R1 and R2 to $1/WC$ (water content) was previously developed by

Fullerton et. al. (1982). However, in our work we used the linear relationship of qMRI parameters with MTV ($=1-WC$). This is due to the fact that such relationship stems from the concept of relaxivity which can be applied to all qMRI parameters and not just to relaxation rates. Interestingly, in the physiological range of MTV values, $1/WC$ varies almost linearly with MTV (see the attached figure).

We've added a supplementary analysis showing that our results remain similar when evaluating the

dependency of qMRI parameters on $1/WC$ (Sup. Fig 30). Following this additional analysis we adjusted the discussion (lines 332-338): *"A theoretical formulation for the effect of the surface interaction on proton relaxation has been proposed before^{75,76}. Specifically, a biophysical model for the linear relationship between $R1$ and $R2$ to the inverse of the water content ($1/WC=1/(1-MTV)$) was suggested by Fullerton et. al.⁶⁰. Interestingly, $1/WC$ varies almost linearly with MTV in the physiological range of MTV values. Applying our approach with $1/WC$ instead of MTV produces relatively similar results (Sup. Fig 30). However, the interpretation of the dependency on MTV as tissue relaxivity allowed us to generalize the linear model to multiple qMRI parameters, thus producing multidimensional MDM signatures"*.

The correlation between MDM and gene expression patterns (Fig 3E) is not very convincing, as also shown by the R^2 of 0.3. I am not suggesting to remove it completely but the Authors should mitigate their claims relative to this particular result, which could be driven by few outliers.

We thank the reviewer for this comment. We believe that a statistically significant model that explains ~30% of the variance between two very different modalities is not neglectable. We want to stress the fact that all the correlations included in the gene expression analysis (57) were corrected for multiple comparisons using the FDR method. Following the reviewer's concerns, **we performed an outlier analysis to the genetics results** (Sup. Fig. 12). Notably, we excluded seven outliers and the significant correlations between the MDM and the gene modules survived. Nevertheless, we agree with Reviewer 2 that this point is less constrictive to the main point of the article. We therefore moved the gene expression analysis to the supplementary and shortened the discussion regarding it.

The TR of the MTsat acquisition in humans seems very long - check this is the correct value. We thank Reviewer 2 for this comment. It was indeed an incorrect value; the true TR we used was shorter (27 ms). We corrected this typo in the methods section.

Reviewer #3:

The authors used 3 quantitative MRI parameters ($R1$, $R2$, MTS_{at}) to estimate lipid and water content of the brain and relate this to changes attributed with age. They used phantoms with different lipid contents, literature values from PM studies for lipid content of different brain regions, literature data on some genes in different cortical regions, and data from 23 younger and 18 older humans. There is substantial interest in characterising brain molecular changes with ageing non-invasively and MRI is a major tool for this

We thank Reviewer 3 for acknowledging the interest in measuring aging-related molecular changes non-invasively. We would like to point out that for the human data we used four quantitative MRI parameters ($R1$, $R2$, MTS_{at} and MD). We are sorry it wasn't clear enough. We highlighted this in the revised manuscript.

However, there are problems with the present approach which limit the interpretation and reliability of the results. There are other quantitative MRI techniques which show sensitive and specific age related changes - eg several DTI parameters, T1 mapping for brain water, as

well as volume loss and lesion formation. The present paper hardly mentioned these and did not compare their multi-sequence method with other more researched approaches.

We thank the reviewer for stressing the importance of the comparison between our novel approach and other conventional MRI techniques. We are well aware of the earlier MRI work regarding brain aging. For example, the first joint T1 and DTI work that shows white matter changes with age was published by the last author of this paper (Yeatman, Nature Communications 2014).

Apparently, the relevant results were not highlighted enough, but **a considerable part of our work deals with comparisons of our method to other aging makers (including T1, DTI and volume loss as the reviewer suggested)**. We therefore completely agree that such comparative analysis is important.

First, we compare the pattern of differentiation between brain regions revealed by the MDM method (Fig. 2C & Sup. Fig 6) to the pattern of differentiation revealed by standard qMRI parameters (R1 (=1/T1), MTsat, R2, and the DTI measure MD, Sup. Fig. 8). Next, we show that the correlation of MRI measurements with histology is enhanced when using the MDM signatures relative to standard qMRI parameters (Fig. 3B relative to Sup. Fig. 9B-C, Fig. 4D,E,H).

We also highlight the differences between our method and other quantitative MRI techniques in the aging analysis. Specifically, in figures 5-6 and sup. figures 14-18 we show that the changes in MRI measurements (two of them are DTI and T1 that the reviewer mentioned) observed with aging result from a combination of alterations in the molecular composition of the tissue (estimated by MDM) and its water content (estimated by MTV).

Importantly, in figure 7 we compare the trajectories of different aging markers throughout the brain. We show that volumetric measurements, iron, water content and our molecular signatures all have distinct spatial patterns of aging related changes. Therefore, only by combining different aging markers a full picture of the aging mosaic can be achieved. In that sense our molecular signatures contribute additional novel dimension that describes molecular aging-related changes independent from the other established markers. This is one of the main take-home messages our work offers. Following the reviewer's comment, we adjusted the manuscript, emphasizing the comparison of MDM to common aging markers. For example, in the conclusion (lines 373-377): *"Results obtained by this novel method in-vivo disentangle different biological process occurring in the brain during aging. Such processes are blended together when using conventional qMRI parameters as aging markers. We identified region-specific patterns of microscale aging-related changes that are associated with the chemophysical composition of the human brain. These changes are independent from other widely established aging markers such as tissue volume alterations"*.

The human subjects are not described at all - were these normal volunteers? What were their medical histories? How were they recruited? Were the young and old groups matched for premorbid IQ which has a major effect on white matter and cortical parameters? Were they matched for socioeconomic status, which also affects brain structure. Other variables such as educational attainment are important and should be balanced between the groups. If the groups were not balanced for these key factors, then other sources of variation between the

individuals could account for observed differences, not age. The lack of characterisation of the humans is a major shortcoming in a paper that purports to present variables that are makers of age.

We thank the reviewer for raising this concern. We agree that it is important to better describe the human subjects. We recruited healthy volunteers from the community surrounding the Hebrew University of Jerusalem. The vast majority of the younger subjects were graduate students, and most of the older subjects were either faculty members in the Hebrew University or working on campus. We therefore believe that variables such as educational attainment were fairly balanced between the age groups. Moreover, our in-vivo aging results are in agreement with previous histological aging studies. Similar to the trend presented in our paper, previous publications reported that the cortex shrinks with age, while the neural density remains relatively constant (Burke, Nature Reviews Neuroscience 2006). Moreover, both our research and previous histological studies found significant molecular alterations in the white matter during aging (Bowley, The Journal of Comparative Neurology 2010). In addition, we find that the shrinkage of the hippocampus with age is accompanied with conserved tissue density and chemophysical composition. This is in agreement with histological findings, which predict drastic changes in hippocampal tissue composition in neurological diseases such as Alzheimer, but not in normal aging (Freeman, Journal of Neuropathology & Experimental Neurology 2008). Still, it is true that other sources of variation between individuals can explain the observed differences between the groups. Following this comment, we adjusted the methods section: *"Human measurements were performed on 23 young adults (aged 27 ± 2 years, 11 females), and 18 older adults (aged 67 ± 6 years, 5 females). Healthy volunteers were recruited from the community surrounding the Hebrew University of Jerusalem"*.

In addition, we addressed this concern in the discussion (lines 359-363): *"Most of the human subjects were recruited for this study from the academic community. However, it could be that some variables such as IQ and socioeconomic status were not completely balanced between the age groups. While this implies that other sources except age can explain the differences between the groups, we believe that the agreement of our findings with previous histological aging studies supports the association between the group differences we measured and brain aging"*.

The phantom data have limitations - while several different lipids and concentrations were tested, phantoms cannot represent the complexity of human tissue, including not just the concentration of particular molecules but also their binding state, etc.

We thank the reviewer for this comment which was also raised by another reviewer.

Following this comment, **we adjusted the manuscript and performed additional phantom experiments**. As we answered reviewer 1:

We are sorry it wasn't clear enough in the manuscript, but the goal of our phantoms was not to mimic the full complexity of brain tissue. For this aim we applied our method for the in-vivo human brain and compared it to postmortem data. Nevertheless, we performed the phantom experiments in order to verify that our relaxivity method captures information regarding the chemophysical environment. Particularly, the phantom experiment shows examples in which standard qMRI parameters cannot separate between changes in water

fraction and molecular composition (Fig 1A insert). We show this ambiguity can be resolved using the MDM approach (Fig 1X). It is the simplicity of the phantoms that allowed us to prove the sensitivity of our method to very specific molecular components. Using the phantom system, we were able to show the sensitivity of our method to different brain lipids. Moreover, by evaluating mixtures of several lipids, we showed that the composition of a mixture can be predicted using our method. Such accurate predictions of molecular environment with MRI were never shown before. Finally, we show that the same model of prediction can be applied for the human brain to generate accurate predictions of specific molecular features (Fig. 3C).

Following the reviewer's comment, we highlighted this point in the results section (lines 68-70,94-98): *"Using this (phantom) system, we tested whether accounting for the effect of the water content on qMRI parameters will provide sensitivity to fine molecular details such as the head groups that distinguish different membrane phospholipids... The lipids samples do not simulate the full complexity of biological tissues. However, they allow us to estimate the sensitivity of our MRI approach to subtle chemophysical differences between phospholipids, in a completely controlled environment. Moreover, our approach can be applied to other compounds to reveal differences in the MRI signal between different proteins, sugars and ions (Sup. Fig. 1D)"*

Further clarification was added to the discussion (lines 269-275,280-282): *"We developed a unique phantom system of lipid samples to validate our method... Many previous publications assume that MRI measurements are sensitive to different cellular components such as myelin. Our work tests such assumptions on a completely controlled phantom environment in a rigorous manner. While the phantom system is clearly simplistic compared to brain tissue, its simplicity allowed us to verify the specificity of our method to the chemophysical environment. Remarkably, our approach revealed unique signatures for different lipids, and is therefore sensitive even to the relatively subtle details that distinguish one lipid from another... Therefore, the phantoms were made to examine the MRI sensitivity for the chemophysical environment, and the human brain data was used to measure the true biological effects in a complex environment"*.

Moreover, we have performed several additional phantom experiments to demonstrate that the MDM approach can be generalized. We show that sugars, proteins and ions can influence relaxivity and be distinguished using the MDM signatures (Sup. Fig. 1D, for text see lines 94-98). We highlight this point in the discussion (lines 275-280): *"We chose to validate our approach using membrane lipids based on previous NMR experiments. Nevertheless, we acknowledge the fact that brain tissue is comprised of many other compounds beside lipids, such as proteins, sugars and ions. As we have shown, these other compounds also exhibit unique dependency on MTV, thus demonstrating the comprehensive nature of our framework. The effect of such compounds is probably captured when we apply the MDM approach to the in-vivo human brain"*.

Use of literature values for regional brain lipids are a serious limitation. Were the papers on human brain? Were there any details about the brains that provided the samples described in the literature? How many brains contributed and what ages were they?

We thank the reviewer for suggesting to better describe the histological data taken from the literature. **We now describe the histological data taken from the literature in the methods section:** *"The data was taken from 8 post-mortem human brains (Söderberg, 1990). Brains were obtained from individuals between 54-57 years of age, which were autopsied within 24 h after death". Söderberg et. al. note that: "Great care was taken to select tissues which, upon careful histological investigation, did not demonstrate morphologically detectable signs of disease". Indeed, we personally contacted Söderberg and made sure we understand the procedure of the histological experiments. To the extent of our knowledge, this is the most comprehensive publicly available data regarding the lipid composition of the human brain.*

Moreover, we added a clarification in the head of the figure 3 caption: *"Comparison of the in-vivo MDM signatures of different brain regions to the molecular composition of these regions as reported in the literature for 8 post-mortem human brains (Söderberg, 1990)".*

Furthermore, we want to stress the validity of the comparison we made (Fig. 3). Even though our data was averaged over 8 human brains for the histological measurements and 23 human brains for the MRI signatures, the intersubject variabilities are much smaller than the regional variability. Note that in figure 3B the error bars are small relative the range of variability between brain areas. Therefore, despite the fact the our histological and in-vivo measurements were not taken from the same tissue, these measurements are highly stable across normal subjects and the astonishing agreement between the two modalities ($R^2=0.84$) provides a strong evidence for the ability of our method to capture molecular information. We would like to point out that using standard qMRI parameters instead of our new measurements yielded much lower agreement between histological and in-vivo measures (sup. figure 9B-C). Following the reviewer's comment, we adapted the discussion (lines 298-304).

Why not study PM human brain tissue itself with MRI then histology? This would be much more relevant, so that the MRI signatures could be compared directly with the relevant specimen in a PM MRI-histology protocol were used. Several have been developed by other groups. This enables a direct tissue to tissue comparison to be done.

We thank the reviewer for this suggestion which was also raised by reviewer 1. We agree with the reviewer that while lipidomics post-mortem analysis is very challenging there is great importance in performing histological analysis and MRI scans on the same brain. Following these comments, **we made a direct comparison of MRI measurements and histological analysis on post-mortem brains.** To our knowledge this is the first post-mortem analysis that compares between lipidomics and MRI. For this aim we performed MRI scans (R1, MTsat, R2, MD and MTV mapping) followed by thin-layer chromatography (TLC) technique of two fresh post-mortem porcine brains. Even though there are many challenges in scanning post-mortem tissue, segmenting it, and comparing it to anatomically relevant histological result, we were able to replicate our findings in the level of the single brain. First, MTV values estimated using MRI were in agreement with the non-water fraction found histologically (adjusted $R^2=0.64$, $p<0.001$). Moreover, while MTV and standard qMRI parameters could not explain the lipidomics variability in the brain, the MDM signatures correlated significantly with the lipid composition (adjusted $R^2=0.3$, $p<0.01$).

Excluding 5 brain regions (out of 30) with extreme TLC values yielded even stronger correlation between MDM signatures and molecular composition (adjusted $R^2=0.55$, $p<0.001$). The full analysis has been incorporated in the manuscript (Fig. 4, and without outliers in Sup. Fig. 10C). We believe it greatly strengthens the credibility of our approach and demonstrates its specificity to brain lipids. In addition, we provide for the first-time histological validation for the MRI estimation of MTV on brain tissue.

The genetics - there are thousands of genes in the human brain that have associations with a range of diseases of aging - the present analysis was barely described and made little contribution to the reliability of the proposed quantitative MRI markers.

We thank the reviewer for this comment. We would like to clarify that in the gene expression analysis data was taken from a widespread survey of gene expression undertaken by the Allen Human Brain Atlas project (<http://www.brain-map.org>). The thousands of genes in the dataset were clustered into 19 gene families using a WGCNA approach (Ben-David, PLOS genetics 2012). This approach allowed us to compare MRI measures to the spatial expression patterns of a large number of genes represented by the gene modules.

We believe that this analysis contributes to the association of the MDM signatures with the cellular molecular microenvironment. Compared to standard quantitative MRI markers, the MDM signatures correlated significantly with more gene modules (Sup. Fig. 11). Moreover, the correlations of the MDM signatures with the membrane and synapse modules were the strongest among all 57 correlations tested (after correcting for multiple comparisons). In addition, **we incorporated an outlier analysis to the genetics results** to verify they are not driven by a few outliers (Sup. Fig. 12). Notably, we excluded seven outliers and the significant correlation between the MDM and the gene modules survived.

Therefore, the gene expression analysis indicates that the MDM approach enhances the consistency between MRI-driven measures and histology over standard qMRI measurements. Nevertheless, following this concern (and the one of Reviewer 2), we moved the gene expression analysis to the supplementary where we describe it more thoroughly.

Some of the background information was rather overstated. For example, there are many studies ageing in the brain, or diseases that increase with age or accelerate aging, that do not support the concept of 'MRI changes of aging that are usually attributed to myelin' - what is this? Myelin is one of the last components of the brain to change with ageing or indeed in several pathologies such as age related white matter hyperintensities. Advanced quantitative MRI clearly shows that these areas represent increased water content at the earliest stage and even chronically. Several studies show that increase in tissue water content is one of the earliest changes with age, which with other pathologies eventually leads to myelin damage, axon and myelin loss - the idea that myelin damage or loss is an early feature of ageing reflects a longstanding artefact of tissue pathology - namely that the tissue rarefaction is due to water increase not myelin loss, but the water is routinely removed in the histopathology preparation therefore unobserved.

We thank the reviewer for commenting on the fact that changes in tissue water content, and not myelin, are one of the earliest aging markers. The whole purpose of our method is to

separate the effects of water and molecular composition on qMRI parameters. Therefore, we fully agree with Reviewer 3 regarding the importance of distinguishing water and other (e.g. myelin) changes with age. It is important to note that our group developed qMRI techniques for measuring water content and published several studies regarding the importance of this measure in the brain (Mezer et. al., Nature Medicine 2013, Filo & Mezer, PD: Proton Density of Tissue Water in: Quantitative MRI of the Brain). However, this is not the prevalent dogma in the quantitative MRI field. In aging studies, quantitative MRI measurements are more often attributed to myelin than to water content and cannot disentangle these two important biological factors (Heath, Developmental neurobiology 2018, Callaghan, Neurobiology of Aging 2014, Steiger, The Journal of neuroscience 2016). We are sorry it wasn't clear enough in the text, but we now adjusted the sentence quoted by the reviewer (line 45): *"We hypothesized that the changes observed with aging in MRI measurements such as R1, R2, Mean Diffusivity (MD) and Magnetization Transfer Saturation⁵³ (MTsat)^{35,39,45-50,54,55}, could be due to a combination of an increase in water content at the expense of tissue loss, and molecular alterations in the tissue"*.

We could not agree more with Reviewer 3 regarding the importance of the water content in the brain. This is actually the fundamental novelty in our work. The reviewer mentions several studies which show that increase in tissue water content is one of the earliest changes with age, that leads with other pathologies eventually to myelin damage, axon and myelin loss. We would be happy if the reviewer could guide us to this body of work which will further support our argument regarding the importance of separated MRI measurements for water and molecular changes with age. Our method offers for the first time the ability to track the two different trajectories of water and molecular aging related changes in-vivo (see figure 5,6,7). We believe that this information will be invaluable in differentiating normal and abnormal aging in the future.

REVIEWERS' COMMENTS:

Reviewer #1 (Remarks to the Author):

The authors have done an outstanding job responding to the concerns raised by myself and the other reviewer. Their new data bolster their conclusions and the validity of this new and potentially important MRI method.

Reviewer #2 (Remarks to the Author):

The Authors have done extensive work to address the Reviewers' comments, and I find the manuscript greatly improved since first submission.

At this stage I only have minor suggestions to improve readability.

As the paper contains several experiments, it might be useful to very briefly summarise them (in a couple of sentences) at the beginning of the Results section.

Next - when commenting on the human in vivo results - it is important to treat the comparison between age groups as a proof of concept rather than a definite conclusion on changes induced by ageing. Considering that the samples were relatively small and potentially affected by additional factors, it is relevant to stress that these results show the potential of MDM but might not be the ultimate answer to the question "what changes occur in the brain with ageing?".

point-by-point response to the reviewers

Thank you for reviewing our manuscript "Disentangling molecular alterations from water-content changes in the aging human brain using quantitative MRI" (NCOMMS-18-32563A-Z). We have adjusted our manuscript following your requests. The requests are in blue and our responses are in black.

Reviewer #1 (Remarks to the Author):

The authors have done an outstanding job responding to the concerns raised by myself and the other reviewer. Their new data bolster their conclusions and the validity of this new and potentially important MRI method.

We thank Reviewer 1 for agreeing to review our manuscript again and acknowledging the importance of our work.

Reviewer #2 (Remarks to the Author):

The Authors have done extensive work to address the Reviewers' comments, and I find the manuscript greatly improved since first submission.

We thank Reviewer 2 for agreeing to review our manuscript again.

At this stage I only have minor suggestions to improve readability.

As the paper contains several experiments, it might be useful to very briefly summarise them (in a couple of sentences) at the beginning of the Results section.

We agree with the reviewer that it will be useful to highlight the different experiments in the Results section. Following an editorial request, this section is now divided into subsections with informative titles. Due to the need to shorten the manuscript, we believe this could be an appropriate solution that indicates the different experiments done in each section.

Next - when commenting on the human in vivo results - it is important to treat the comparison between age groups as a proof of concept rather than a definite conclusion on changes induced by ageing. Considering that the samples were relatively small and potentially affected by additional factors, it is relevant to stress that these results show the potential of MDM but might not be the ultimate answer to the question "what changes occur in the brain with ageing?".

We thank Reviewer 2 for this comment. We now further stress this issue in the discussion (Lines 326-331): "It should be noted that most of the human subjects recruited for this study were from the academic community. However, the different age groups were not matched for variables such as IQ and socioeconomic status. **In addition, the sample size in our study was quite small. Therefore, the comparison we made between the two age groups may be affected by variables other than age. Our approach may benefit from validation based on larger quantitative MRI datasets^{27,61}.** Yet, we believe we have demonstrated the potential of our method to reveal molecular alterations in the brain."